# iHyperTime: Interpretable Time Series Generation with Implicit Neural Representations

**Elizabeth Fons**                                              *elizabeth.fons@jpmorgan.com*
*J.P. Morgan AI Research*

**Alejandro Sztrajman**                               *alejandro.sztrajman@cl.cam.ac.uk*
*University of Cambridge*

**Yousef El-Laham**                                           *yousef.ellaham@jpmorgan.com*
*J.P. Morgan AI Research*

**Andrea Coletta**                                           *andrea.coletta@bancaditalia.it*
*Bank of Italy*

**Alexandros Iosifidis**                                                        *ai@ece.au.dk*
*Aarhus University*

**Svitlana Vyetrenko**                                    *svitlana.vyetrenko@jpmorgan.com*
*J.P. Morgan AI Research*

**Reviewed on OpenReview:** *https://openreview.net/forum?id=GSnGPgeoS5*

## Abstract

Implicit neural representations (INRs) have emerged as a powerful tool that provides an accurate and resolution-independent encoding of data. Their robustness as general approximators has been shown across diverse data modalities, such as images, video, audio, and 3D scenes. However, little attention has been given to leveraging these architectures for time series data. Addressing this gap, we propose an approach for time series generation based on two novel architectures: TSNet, an INR network for interpretable trend-seasonality time series representation, and iHyperTime, a hypernetwork architecture that leverages TSNet for time series generalization and synthesis. Through evaluations of *fidelity* and *usefulness* metrics, we demonstrate that iHyperTime outperforms current state-of-the-art methods in challenging scenarios that involve long or irregularly sampled time series, while performing on par on regularly sampled data. Furthermore, we showcase iHyperTime fast training speed, comparable to the fastest existing methods for short sequences and significantly superior for longer ones. Finally, we empirically validate the quality of the model's unsupervised trend-seasonality decomposition by comparing against the well-established STL method.

## 1 Introduction

Modeling time series data has been a key topic of research for many years, constituting a crucial component in a wide variety of areas such as climate modeling, medicine, biology, retail and finance (Lim & Zohren, 2021). Traditional methods for time series modeling have relied on parametric models informed by expert knowledge.

However, the development of modern machine learning methods has provided purely data-driven techniques to learn temporal relationships. In particular, neural network-based methods have gained popularity in recent times, with applications to a wide range of tasks, such as time series classification (Ismail Fawaz et al., 2020), clustering (Alqahtani et al., 2021), segmentation (Zeng et al., 2022), anomaly detection (Choi et al., 2021), upsampling (Oh et al., 2020), imputation (Cao et al., 2018; Shukla & Marlin, 2021), forecasting (Torres et al., 2021) and generation (Coletta et al., 2023). In particular, generation of synthetic time series has recently gained attention due to the large number of applications in medical and financial fields, where data cannot be shared, either due to privacy reasons or proprietary restrictions (Jordon et al., 2021; Assefa et al., 2020). Moreover, synthetic time series can be used to augment training datasets to improve model generalization on downstream tasks, such as classification (Fons et al., 2021), forecasting and anomaly detection. In these fields, having a disentangled representation of time series can be critical for applications with regulatory focus, which often require transparency and interpretability of proposed machine learning solutions as well as injection of expert knowledge as constraints into the training process (Vyetrenko & Xu, 2019).

The task of generating realistic time-series data poses considerable challenges, particularly due to the diverse nature of time series, which may vary along dimensions such as univariate vs. multivariate, short vs. long, and regularly vs. irregularly sampled. Although numerous solutions have been proposed to address the problem of time-series data generation (Alaa et al., 2021; Yoon et al., 2019; Esteban et al., 2017), the focus has predominantly been on generating data for well-structured scenarios, such as short and regularly sampled time series. Such an approach is often conflicting with the complexities of real-world time-series data, where irregularities, missing values and diverse sequence lengths are commonplace (Fang & Wang, 2020). Recent methods have addressed some of these shortcomings (Jeon et al., 2022; Zhou et al., 2023; Coletta et al., 2023). However, no single approach has shown a consistent performance across all these challenging scenarios. Furthermore, the scalability of many existing methods to longer sequences is constrained by escalating computational costs that correlate with series length.

In recent years, implicit neural representations (INRs) have gained popularity as an accurate and flexible method to parameterize signals from diverse sources, such as images, video, audio and 3D scene data (Sitzmann et al., 2020b; Mildenhall et al., 2020). Conventional methods for data encoding often rely on discrete representations, such as data grids, which are limited by their spatial resolution and present inherent discretization artifacts. In contrast, INRs encode data in terms of continuous functional relationships between signals, and thus are uncoupled to spatial resolution. In practical terms, INRs provide a data representation framework that is resolution-independent, which makes them ideally suited to deal with the aforementioned challenges in real-world time series. While there have been a few recent works exploring the application of INRs to time series data (Jeong & Shin, 2022; Woo et al., 2023), there is no work on leveraging these architectures for generating synthetic time series.

In this paper, we propose a novel method for time series generation based on two novel architectures: 1) TSNet, an INR tailored for resolution-agnostic encoding of time series data, offering a trend-seasonality-residual disentangling of single time series. 2) iHyperTime, a hypernetwork architecture for generalization of time series datasets, that leverages TSNet to produce interpretable latent representations of the signals. Together, these architectures form a unified approach for disentangled representation and generation of multiple forms of time series data, including challenging cases such as multivariate, irregularly sampled, and long time series.

**Generation quality** Through empirical evaluations, we demonstrate that iHyperTime outperforms existing state-of-the-art methods for time series generation. Our method excels in complex scenarios such as irregularly sampled or long sequences, while performing on par with state-of-the-art for regularly sampled time series.

**Efficiency** We show that our architecture achieves rapid training speeds, comparable to the fastest methods for short sequences, and substantially faster for longer ones.

**TSR Decomposition** We validate the unsupervised decomposition capabilities of our method by benchmarking it against the widely-used STL decomposition technique.

## 2    Related Work

**Implicit Neural Representations**   Implicit Neural Representations (INRs) provide a continuous representation of multidimensional data, by encoding a functional relationship between input coordinates and signal values, avoiding possible discretization artifacts. They have recently gained popularity in visual computing (Mescheder et al., 2019; Mildenhall et al., 2020) due to the key development of positional encodings (Tancik et al., 2020) and SIREN periodic activations (Sitzmann et al., 2020b), which have proven to be critical for the learning of high-frequency details. Whilst INRs have been shown to produce accurate reconstructions in a wide variety of data sources, such as video, images and audio (Sitzmann et al., 2020b; Chen et al., 2021; Rott Shaham et al., 2021), few works have leveraged them for time series representation (Jeong & Shin, 2022; Woo et al., 2023), and none have focused on interpretability and generation.

**Hypernetworks**   Hypernetworks are neural network architectures that are trained to predict the parameters of secondary networks, referred to as hyponetworks (Ha et al., 2017; Sitzmann et al., 2020a). In the last few years, some works have leveraged different hypernetwork architectures for the prediction of INR weights, in order to learn priors over image data (Skorokhodov et al., 2021) and 3D scene data (Littwin & Wolf, 2019; Sitzmann et al., 2019; Sztrajman et al., 2021). Sitzmann et al. (2020b) leverage a set encoder and a hypernetwork decoder to learn a prior over SIRENs encoding image data, and apply it for image in-painting.

**Time Series Generation**   Synthesis of time series data using deep generative models has been previously studied in the literature. Examples include the TimeGAN architecture (Yoon et al., 2019), GT-GAN (Jeon et al., 2022), and QuantGAN (Wiese et al., 2020). More recently, as an alternative to GAN-based time series generation, Desai et al. (2021) proposed TimeVAE, based on a variational autoencoder, while Coletta et al. (2023) proposed a diffusion model architecture called DiffTime. Alaa et al. (2021) introduced Fourier Flows, a normalizing flow model for time series data that leverages the frequency domain representation, which is currently considered together with TimeGAN as state-of-the-art for time series generation. In the last few years, multiple methods have used INRs for data generation, with applications on image synthesis (Skorokhodov et al., 2021), super-resolution (Chen et al., 2021) and panorama synthesis (Anokhin et al., 2021). However, there are currently no applications of INRs on the generation of time series data.

**Interpretable Time Series**   Seasonal-trend decomposition is a standard tool in time series analysis. The trend encapsulates the slow time-varying behavior of the signal, while seasonal components capture periodicity. These techniques introduce interpretability in time series, which plays an important role in downstream tasks such as forecasting and anomaly detection. The classic approaches for decomposition are the widely used STL algorithm (Cleveland et al., 1990), and its variants (Wen et al., 2019; Bandara et al., 2022). Relevant to this work is the recent N-BEATS architecture (Oreshkin et al., 2020), a deep learning-based model for univariate time series forecasting that provides interpretability capabilities. The model explicitly encodes seasonal-trend decomposition into the network by defining separate trend and seasonal blocks, which fit a low degree polynomial and a Fourier series.

## 3    Formulation

In this Section, we describe the TSNet network architecture for time series representation and TSR decomposition, and the iHyperTime network leveraged for generalization and new data generation.

### 3.1    Time Series Representation

We consider a time series signal encoded by a discrete sequence of $N$ observations $\mathbf{y} = (\mathbf{y}_1, ..., \mathbf{y}_N)$ where $\mathbf{y}_i \in \mathbb{R}^m$ is the $m$-dimensional observation at time $t_i$. This time series defines a dataset $\mathcal{D} = \{(t_i, \mathbf{y}_i)\}_{i=1}^{N}$ of time coordinates $t_i$ associated with observations $\mathbf{y}_i$. We want to find a continuous mapping $f : \mathbb{R} \to \mathbb{R}^m, t \to f(t)$ that parameterizes the discrete time series, so that $\mathbf{y}_i = f(t_i)$ for $i = 1, \ldots, N$. The function $f$ can be approximated by an implicit neural representation (INR) architecture conditioned on the training loss $\mathcal{L} = \sum_i \|\mathbf{y}_i - \hat{f}(t_i)\|^2$. Input and output of the INR are of dimensions 1 and $m$, corresponding to the time coordinate $t$ and the prediction $\hat{f}(t)$. After training, the network encodes a continuous representation of the functional relationship $f(t)$ for a single time series.

### 3.1.1 TSNet

We propose a novel INR architecture, namely TSNet, that allows for interpretable time series representation with TSR decomposition. TSNet encodes a time series as a continuous function as described in the previous section. In particular, we assume that our proposed INR follows a classic time series additive decomposition, i.e.,

$$f(t) = f_T(t) + f_S(t) + f_R(t), \tag{1}$$

where $f_T$, $f_S$, $f_R$ correspond to the trend, seasonality and residual components of $f(t)$. Note that this is a standard assumption for time series decomposition techniques, such as STL and others (Cleveland et al., 1990). We elaborate on our modeling of these three components in the following.

**Trend and Seasonality Blocks**  Following the work by Oreshkin et al. (2020), we model trend and seasonality via basis decompositions with coefficients learned by fully-connected networks. The trend component of a time series aims to model slow-varying (and occasionally monotonic) behavior, thus we consider a polynomial regressor, i.e.,

$$f_T(t) = \sum_{p=0}^{P} \mathbf{w}_p^{(T)} t^p, \tag{2}$$

where $P$ denotes the degree of the polynomial, and $\mathbf{w}_p^{(T)}$ denotes the learned weight associated with the $p$th degree. In practice, $P$ is chosen to be small (e.g., $P = 2$) to capture low frequency behavior.

The seasonal component of the time series $f_S(t)$ aims to capture the periodic behavior of the signal, and thus we leverage a learnable Fourier decomposition:

$$f_S(t) = \sum_{i=0}^{N/2-1} \left( \mathbf{w}_i^{(S)} \cos\left(2\pi it\right) + \mathbf{w}_{N/2+i}^{(S)} \sin\left(2\pi it\right) \right) \tag{3}$$

where $\mathbf{w}_i^{(S)}$ are the weights predicted by the network.

**Residual Block**  The residual of a time series comprises the high-frequency non-periodic components of the signal. In order to model it, we leverage a fully-connected network of $K$ layers with sine activations (SIREN), as defined by Sitzmann et al. (2020b):

$$\mathbf{q}_{k+1} = \sin\left(\omega_0 \mathbf{w}_k^{(R)} \mathbf{q}_k + \mathbf{b}_k^{(R)}\right), \qquad k = 0, ..., K-1 \tag{4}$$

$$f_R(t) = \mathbf{w}_K^{(R)} \mathbf{q}_K + \mathbf{b}_K^{(R)} \tag{5}$$

where $\mathbf{w}_k^{(R)}$, $\mathbf{b}_k^{(R)}$ and $\mathbf{q}_k$ are the weights, biases, and outputs of the $k$ layer, with $\mathbf{q}_0 = t$ corresponding to the input of the network. A general factor $\omega_0$ multiplying the network weights determines the order of magnitude of the frequencies that will be used to encode the signal. As shown by Sitzmann et al. (2020b), SIRENs mitigate the spectral bias of regular fully-connected networks, and thus are well suited for learning and representation of high-frequencies. We refer to Appendix D.2 for the TSNet model implementation details.

In summary, TSNet allows for the decomposition of time series into trend, seasonality, and residual components. While a standard INR, such as a SIREN, can be used to encode a time series as shown in Section 3.1, it would lack the interpretable decomposition that our TSNet INR has. This interpretability is a significant advantage over traditional INRs, which often lack such explicit decomposition capabilities.

## 3.2 Time Series Generation with iHyperTime

In Fig. 1, we show a diagram of our iHyperTime architecture, which can be used to learn a prior over implicit neural representations (TSNet) of time series data. Next, we will detail its components and describe how iHyperTime can be used for time series generation. Additional details on the model implementation can be found in Appendix D.2.

**Set Encoder**  The set encoder is composed of SIREN layers (Sitzmann et al., 2020b) and takes as input an arbitrary set of tuples $\{(t_i, \mathbf{y}_i)\}_{i=1}^{N}$, where $t$ denotes the time-coordinate and $\mathbf{y}_i$ the corresponding univariate

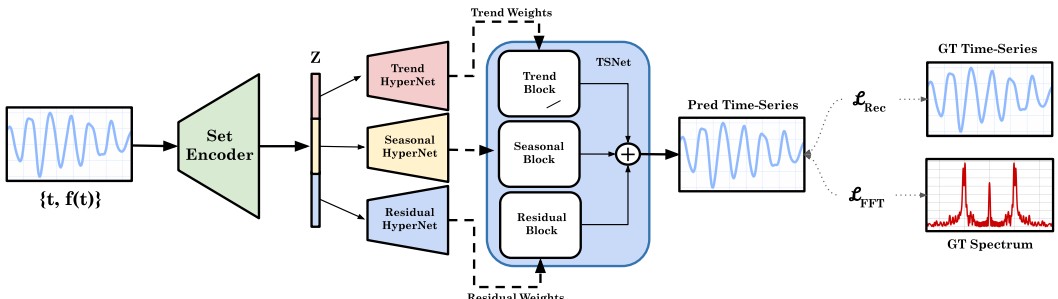

Figure 1: iHyperTime architecture. The Set Encoder processes a set of tuples $\{(t_i, \mathbf{y}_i)\}_{i=1}^N$ representing a time series, and encodes it as embeddings $Z_T$, $Z_S$, $Z_R$ associated to the components of the TSR decomposition. The hypernetwork decoders learn to predict the weights of their corresponding TSNet blocks from the embeddings. During training, the output of the hypernetworks is used to instantiate a TSNet hyponetwork, and the loss is computed as a difference between $\mathbf{y}$ and the output of TSNet $\hat{f}(t)$, in terms of signal and spectral distribution.

or multivariate time series value. Each tuple is encoded into a fixed-size embedding $Z_i = g(t_i, \mathbf{y}_i)$, and the sample set is reduced to a single embedding $Z$ by applying a symmetric operation $\oplus$ (e.g., averaging): $Z = \bigoplus_{i=1}^N Z_i$, where the function $g : \mathbb{R} \times \mathbb{R}^m \to \mathbb{R}^{d_Z}$, used to determine the embedding of each tuple, is parameterized by the SIREN layers (model details in Appendix D.2). The use of a set encoder introduces permutation invariance in the computation, and provides a high degree of flexibility in terms of the input (Zaheer et al., 2017), enabling the encoding of data with irregular sampling, which are common occurrences in time series.

**Hypernetwork decoders** The embedding $Z$ is modeled as a concatenation of three sub-embeddings $Z_T$, $Z_S$, and $Z_R$ with $Z_T \in \mathbb{R}^{d_T}$ denoting the trend embedding, $Z_S \in \mathbb{R}^{d_S}$ denoting the seasonality embedding, and $Z_R \in \mathbb{R}^{d_R}$ denoting the residual embedding with $d_Z = d_T + d_S + d_R$. Each embedding component is pass through its own hypernetwork decoder, which outputs the weights of its corresponding block in the TSNet INR. For example, $Z_T$ is passed into the Trend Hypernetwork to output the weights of the trend block in TSNet. The output of TSNet sums the three signals from each block into a single predicted time series, which is compared against the ground truth signal via the reconstruction and spectral losses ($\mathcal{L}_{Rec}$ and $\mathcal{L}_{FFT}$) during training.

**iHyperTime training** During training, we use the weights predicted by the hypernetwork decoders to instantiate a TSNet hyponetwork and evaluate it on the input time-coordinate $t$, to produce the predicted time series value $\hat{f}(t)$. We then compare the TSNet hyponetwork prediction with the ground truth signal via the reconstruction and spectral losses $\mathcal{L}_{Rec}$ and $\mathcal{L}_{FFT}$.

The training of iHyperTime is performed in three stages, in order to improve stability: 1) we train the Trend networks (Trend hypernetwork, Trend block) for 100 epochs, computing the MSE loss between the ground truth time series $\mathbf{y}$ and the output of the block: $\mathcal{L}_1 = \sum_i \|\mathbf{y}_i - \hat{f}_T(t_i)\|^2$. This leads to a smooth approximation of the time series, which we use as initial guess for the second stage. 2) We then train the Trend and Seasonality blocks together, computing the MSE reconstruction loss $\mathcal{L}_{rec}$ and the FFT loss $\mathcal{L}_{FFT}$ between the ground truth and the added output of both TSNet blocks. 3) Finally, we train the three blocks together.

**Training Loss** The training of iHyperTime is done by optimizing the following loss, which contains an MSE reconstruction term $\mathcal{L}_{rec}$, a spectral loss $\mathcal{L}_{FFT}$ and two regularization terms $\mathcal{L}_{weights}$ and $\mathcal{L}_{latent}$, for the network weights and the latent embeddings, respectively:

$$\mathcal{L} = \underbrace{\frac{1}{N} \sum_{i=1}^N \left\| \mathbf{y}_i - \hat{f}(t_i) \right\|^2}_{\mathcal{L}_{rec}} + \lambda_1 \underbrace{\frac{1}{W} \sum_{j=1}^W w_j^2}_{\mathcal{L}_{weights}} + \lambda_2 \underbrace{\frac{1}{Z} \sum_{l=1}^Z z_l^2}_{\mathcal{L}_{latent}} + \lambda_3 \underbrace{\frac{1}{N} \sum_{k=0}^{N-1} \|F_k - \hat{F}_k\|}_{\mathcal{L}_{FFT}}. \tag{6}$$

where $F_k = [\mathcal{F}_T\{\mathbf{f}\}]_k$ corresponds to the coefficient of the $k$th frequency in the discrete Fourier transform (DFT) of the time series. The term $\mathcal{L}_{\text{FFT}}$ penalizes deviations of the signal's frequency spectrum with respect to ground truth. Thus, we ensure a high-fidelity reconstruction not only of the time series values, but also of its spectral composition. We refer to Appendices C and D for further details on $\mathcal{L}_{\text{FFT}}$ and the implementation details of the iHyperTime architecture.

**Time Series Generation**  After training, we leverage the hypernetwork architecture to generate latent representations of the time series from our training set. Generation of new time series is produced by randomly selecting pairs of time series, and performing linear a interpolation between their embeddings $Z^{(1)}$ and $Z^{(2)}$:

$$Z^{\text{gen}} = Z^{(1)} + \lambda \left( Z^{(2)} - Z^{(1)} \right) \tag{7}$$

where $\lambda$ is also sampled randomly. Optionally, the interpolation can be performed on individual components $Z_T$, $Z_S$, $Z_R$ of the embeddings, enabling the conditional generation of time series.

## 4  Experiments

We present our evaluation of time series generation on regular and irregular data, covering time series of diverse lengths and numbers of channels. Additionally, we perform an analysis of our model's TSR decomposition, and we compare training and inference times with previous works.

### 4.1  Baselines and Evaluation

**Datasets**  We test the performance of iHT using multiple datasets with varying characteristics such as periodicity, level of noise, number of features and length of the series. Stock corresponds to Google stock price data from 2004 to 2019, where each observation has 6 features. Energy is a UCI appliance prediction dataset (Candanedo et al., 2017) with 28 features. Additionally, we also consider Monash dataset (Godahewa et al., 2021), from which we choose FRED-MD, NN5 Daily, Temperature Rain, and Solar Weekly datasets. A complete description of the datasets can be found in Appendix B.

Additionally, we conduct experiments on irregularly sampled time series, achieved by randomly removing fixed percentages of values from each time series. We create the datasets by removing 30, 50 and 70% of each time series.

**Baselines**  We compare our method with TimeGAN (Yoon et al., 2019), GT-GAN (Jeon et al., 2022), Fourier Flows (FF) (Alaa et al., 2021), LS4 (Zhou et al., 2023), DiffTime (Coletta et al., 2023), and RCGAN (Esteban et al., 2017). TimeGAN and GT-GAN have shown strong performance on multivariate time series with short sequence lengths and are able to handle irregularly sampled data. LS4, DiffTime, and Fourier Flows have shown strong performance on time series with longer sequence length, generating distributions of frequencies that closely resemble the original data. We refer to Appendix D.1 for further details on baselines and the adjusted DiffTime and RCGAN architectures, introduced to deal irregular data and longer time series, respectively.

**Evaluation metrics**  To asses the quality of the synthesized data, we adopt the predictive and discriminative scores used in TimeGAN (Yoon et al., 2019). The **predictive score** measures the *usefulness* of the generated data by using a *train on synthetic, test on real* (TSTR) approach: a model is trained using the synthetic data to predict the next step in a sequence, and then it is evaluated using the real data. In the case of multivariate time series with $m$ channels, $m - 1$ channels are used to predict the next step of the $m^{th}$ channel. The mean absolute error (MAE) between the predicted values and the ground truth is used for the evaluation. In the case of irregular time series, and in order to make a fair comparison with other methods for this type of data, we adopt the more challenging predictive task proposed by GT-GAN (Jeon et al., 2022), which involves predicting the entire future vector, instead of a single channel. Irregular time series often have randomly sampled time steps and channels, and predicting only one future element can make the evaluation less reliable. The **discriminative score** serves as a measurement of *fidelity* of the generated data, where the aim is to assess if the synthetic data is indistinguishable from real data. For this purpose, a discriminative model is

trained to classify real and fake samples, and then used to test whether the original and generated data are correctly classified. The discriminative score is computed as $|\text{Accuracy} - 0.5|$, where a low value means that the classification is challenging, and therefore, the model cannot tell which samples are real and which are generated. For a **qualitative evaluation**, we analyze the synthesized and original time series by employing t-SNE visualizations which project the data into a two-dimensional space (van der Maaten & Hinton, 2008). Additionally, we perform a kernel density estimation (Jeon et al., 2022) to compare the data distributions. For the experiments on longer real-world time series from the Monash dataset, we also consider the **marginal score** (Zhou et al., 2023) that computes the absolute difference between the real and synthetic empirical probability density functions.

## 4.2 Experimental results on regular time series synthesis

Performance results for regularly sampled time series are provided in Table 1, for univariate and multivariate datasets of varying lengths. The results indicate that iHT outperforms all methods in terms of the predictive score. This highlights the usefulness of the data generated by our method as a source of synthetic data for learning. In terms of discriminative score, iHT is competitive with state-of-the-art methods across all datasets, although no method emerges as a definitively superior approach. In Figure 2, the t-SNE visualizations show that the time series generated by iHT closely resemble the ground truth data distribution. We refer to Appendix L for additional qualitative comparisons.

Table 1: Regular time series generation performance in terms of predictive and discriminative scores.

|  | Method | Energy24 | Stock24 | Stocks72 | Stock360 |
|---|---|---|---|---|---|
|  | iHT | **.251** ± **.000** | **.037** ± **.000** | **.188** ± **.000** | **.168** ± **.000** |
| Predictive Score | GT-GAN | .321 ± .002 | .040 ± .000 | .207 ± .000 | .188 ± .000 |
|  | TimeGAN | .273 ± .004 | .038 ± .001 | .226 ± .002 | .206 ± .000 |
|  | RCGAN | .292 ± .005 | .040 ± .001 | .192 ± .001 | .189 ± .000 |
|  | DiffTime | .252 ± .000 | .038 ± .001 | .213 ± .000 | .215 ± .000 |
|  | LS4 | .295 ± .001 | .103 ± .001 | .194 ± .000 | **.168** ± **.000** |
|  | FF | **.251** ± **.000** | .076 ± .001 | .191 ± .000 | .169 ± .000 |
|  | Original | .250 ± .003 | .036 ± .001 | .186 ± .001 | .167 ± .001 |
|  | iHT | .245 ± .019 | **.044** ± **.011** | .014 ± .009 | .018 ± .015 |
| Discriminative Score | GT-GAN | **.221** ± **.068** | .077 ± .031 | .058 ± .017 | .085 ± .064 |
|  | TimeGAN | .236 ± .012 | .102 ± .021 | .073 ± .047 | .042 ± .074 |
|  | RCGAN | .336 ± .017 | .196 ± .027 | **.012** ± **.09** | **.014** ± **.007** |
|  | DiffTime | .445 ± .004 | .097 ± .016 | .097 ± .012 | .101 ± .018 |
|  | LS4 | .499 ± .000 | .363 ± .027 | .089 ± .081 | .088 ± .081 |
|  | FF | .499 ± .001 | .349 ± .113 | .016 ± .018 | .015 ± .014 |

Table 2: Irregular time series generation performance: predictive and discriminative scores. 30% missing data.

|  | Method | Energy24 | Stock24 | Stocks72 | Stock360 |
|---|---|---|---|---|---|
|  | iHT | **.049** ± **.001** | **.013** ± **.001** | **.188** ± **.000** | **.168** ± **.000** |
| Pred Score | GT-GAN | .066 ± .001 | .021 ± .003 | .206 ± .000 | .196 ± .000 |
|  | DiffTime | .052 ± .001 | .019 ± .006 | .200 ± .000 | .188 ± .000 |
|  | LS4 | .063 ± .001 | .022 ± .005 | .198 ± .000 | .229 ± .000 |
|  | FF | .148 ± .007 | .137 ± .029 | .210 ± .000 | .184 ± .000 |
|  | Original | .045 ± .001 | .011 ± .002 | .186 ± .001 | .167 ± .001 |
|  | iHT | .452 ± .003 | **.059** ± **.046** | **.017** ± **.007** | **.014** ± **.010** |
| Disc Score | GT-GAN | .333 ± .063 | .251 ± .097 | .068 ± .007 | .111 ± .026 |
|  | DiffTime | **.298** ± **.010** | .215 ± .010 | .110 ± .045 | .057 ± .070 |
|  | LS4 | .500 ± .000 | .495 ± .004 | .203 ± .028 | .067 ± .016 |
|  | FF | .500 ± .000 | .497 ± .005 | .223 ± .092 | .156 ± .102 |

Table 3: Irregular time series generation performance: predictive and discriminative scores. 50% missing data.

|  | Method | Energy24 | Stock24 | Stocks72 | Stock360 |
|---|---|---|---|---|---|
|  | **iHT (Ours)** | **.051** ± **.002** | **.014** ± **.001** | **.187** ± **.000** | **.168** ± **.000** |
| Pred Score | GT-GAN | .064 ± .001 | .018 ± .002 | .195 ± .000 | .195 ±.000 |
|  | DiffTime | .057 ± .001 | .024 ± .002 | .278 ± .000 | .186 ± .000 |
|  | LS4 | .065 ± .002 | .033 ± .005 | .212 ± .000 | .197 ± .000 |
|  | FF | .227 ± .004 | .169 ± .018 | .236 ± .000 | .215 ± .000 |
|  | Original | .045 ± .001 | .011 ± .002 | .186 ± .001 | .167 ± .001 |
|  | **iHT (Ours)** | .472 ± .004 | **.102** ± **.051** | **.011** ± **.003** | **.004** ± **.003** |
| Disc Score | GT-GAN | **.317** ± **.010** | .265 ± .073 | .026 ± .012 | .081 ± .023 |
|  | DiffTime | .422 ± .011 | .332 ± .034 | .284 ± .137 | .110 ± .061 |
|  | LS4 | .500 ± .000 | .498 ± .000 | .144 ± .034 | .027 ± .015 |
|  | FF | .500 ± .002 | .498 ± .003 | .376 ± .130 | .422 ± .058 |

Table 4: Irregular time series generation performance: predictive and discriminative scores. 70% missing data.

|  | Method | Energy24 | Stock24 | Stocks72 | Stock360 |
|---|---|---|---|---|---|
|  | **iHT (Ours)** | **.053** ± **.000** | **.014** ± **.013** | **.187** ± **.000** | **.168** ± **.000** |
| Pred Score | GT-GAN | .076 ± .001 | .020 ± .005 | .205 ± .000 | .196 ± .000 |
|  | LS4 | .084 ± .003 | .024 ± .002 | .188 ± .000 | .188 ± .000 |
|  | DiffTime | .065 ± .001 | .068 ± .063 | .284 ± .000 | .196 ± .000 |
|  | FF | .304 ± .005 | .205 ± .001 | .267 ± .000 | .245 ± .000 |
|  | Original | .045 ± .001 | .011 ± .002 | .186 ± .001 | .167 ± .001 |
|  | **iHT (Ours)** | .482 ± .003 | **.115** ± **.052** | **.020** ± **.019** | **.011** ± **.012** |
| Disc Score | GT-GAN | **.325** ± **.047** | .230 ± .053 | .058 ± .002 | .091± .013 |
|  | LS4 | .499 ± .002 | .455 ± .011 | .036 ± .026 | .183 ± .017 |
|  | DiffTime | .444 ± .001 | .421 ± .003 | .436 ± .009 | .148 ± .018 |
|  | FF | .500 ± .003 | .498 ± .008 | .424 ± .083 | .369 ± .163 |

## 4.3 Experimental results on irregular time series synthesis

Tables 2, 3 and 4 show the results for the irregular time series generation with different percentages of dropped values. iHT outperforms all methods in terms of predictive score across all datasets. Furthermore, the

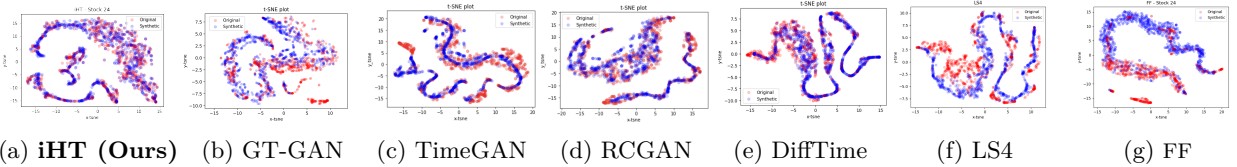

(a) **iHT (Ours)**  (b) GT-GAN  (c) TimeGAN  (d) RCGAN  (e) DiffTime  (f) LS4  (g) FF

Figure 2: t-SNE visualizations on Stock24 data, where a greater overlap of blue and red dots shows a better distributional-similarity between the original and generated data. Our approach shows the best performance. (See Appendix 25 for high resolution charts)

performance of iHT does not degrade significantly with the percentage of dropped values, even in the extreme case of 70% missing data, demonstrating its usefulness in downstream tasks. Given that for irregular data we adopt the predictive score proposed by GTGAN (Jeong & Shin, 2022), which predicts the future values of all channels instead of a single one, we see a drop in predictive score with regards to the values presented in Table 1. In the case of Energy24, which contains 28 channels, the more challenging task of predicting all channels instead of just one may result in a lower predictive score (better performance) compared to the TimeGAN evaluation. This is because the error is averaged over all 28 channels, potentially masking poor performance on individual channels. This effect is also observed, although to a lesser extent, in the Stock24 dataset, which contains 6 channels. For the univariate datasets (Stock72 and Stock360), the predictive score remains largely unchanged as there are no additional channels to leverage. In regard to *fidelity* (discriminative score), our method shows the best performance in 3 out of 4 datasets. Specifically, iHT excels on long time series datasets, demonstrating its ability to handle sequences beyond 24 time steps, even with high percentages of missing data. For the shorter datasets, it shows a degradation in discriminative score when the percentage of dropped data increases. The top row in Figure 3 compares the distributions of original and synthetic data for the Stock24 dataset with 50% of removed values. iHT shows the best performance, with the closest match to the original data distribution. The bottom row shows the corresponding t-SNE visualizations, where we can see that iHT and GT-GAN show the best overlap between original and generated data, with iHT showing more data diversity, covering a wider area across the original data. Additional plots for other missing rates are shown in Appendix L, where we observe a similar behavior with iHT showing the best overlap in both distribution and t-SNE visualization.

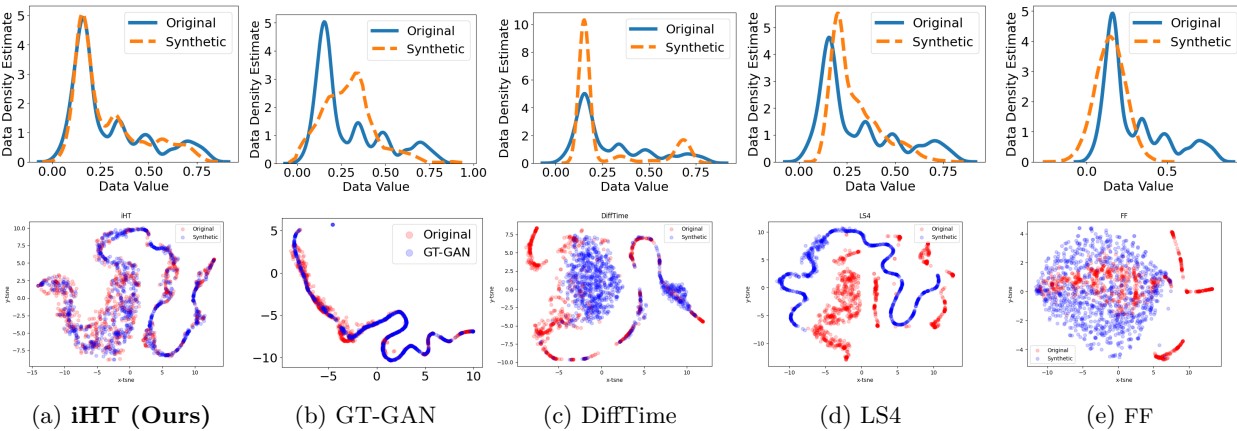

(a) **iHT (Ours)**    (b) GT-GAN    (c) DiffTime    (d) LS4    (e) FF

Figure 3: (Top) Data distribution on irregular Stock24 data (Missing 50%). (Bottom) t-SNE visualizations on irregular Stock24 data (Missing 50%), where a greater overlap of blue and red dots shows a better distributional-similarity between the generated data and original data. Our approach shows the best performance.

## 4.4 Experimental results on longer real-world time series synthesis

Table 5 shows additional generation results for 4 real-world datasets from the Monash dataset, where 3 of the datasets contain time series with lengths over 700 time steps. We report comparisons with LS4, which has

shown state-of-art performance on these datasets (Zhou et al., 2023), while we leave the full evaluation table in Appendix E. In this scenario, the Classification (*fidelity*) and Prediction (*usefulness*) scores are computed using a 1-layer S4 model (Zhou et al., 2023). The results show the ability of iHT to deal with long time series, with superior performance in 3 out of 4 datasets w.r.t. the state-of-art LS4.

Table 5: Generation results on Monash datasets.

| Data | Metric | LS4 | iHT (Ours) | Data | Metric | LS4 | iHT (Ours) |
|---|---|---|---|---|---|---|---|
| FRED-MD | Marginal ↓ | 0.0221 | **0.0177** | Temp Rain | Marginal ↓ | **0.0834** | 0.2978 |
| | Class ↑ | 0.544 | **1.3278** | | Class ↑ | 0.976 | **11.2493** |
| | Prediction ↓ | 0.0373 | **0.0181** | | Prediction ↓ | 0.521 | **0.132** |
| NN5 Daily | Marginal ↓ | **0.00671** | 0.00893 | Solar Weekly | Marginal ↓ | 0.0459 | **0.03273** |
| | Class ↑ | **0.636** | 0.4982 | | Class ↑ | 0.683 | **1.2413** |
| | Prediction ↓ | 0.241 | **0.2349** | | Prediction ↓ | 0.141 | **0.0739** |

## 4.5 Runtime

In Figure 4, we show that the strong performance of iHT on long time series does not impact its computational time. We consider a set of synthetic datasets with lengths {80, 320, 1280, 5120, 20480} and we evaluate the training time for 100 iterations, and the inference time on one batch (Zhou et al., 2023). The figure shows that iHT has among the lowest computational times w.r.t. existing approaches. Moreover, iHT training times are almost unaffected by the length of the time series, with negligible changes even for 20,480 time steps, making it the fastest method for long sequences. Additional details and the overall training times are presented in Appendix G.

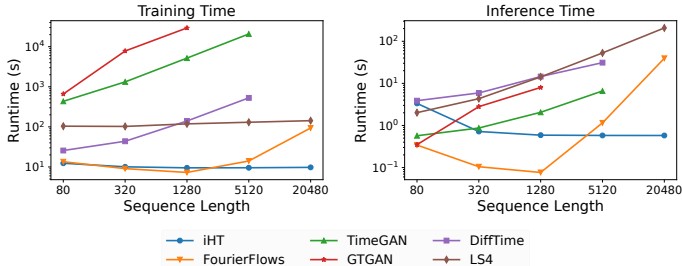

Figure 4: Training and inference time comparison for time series of different lengths.

## 4.6 Trend-Seasonality decomposition analysis

iHT provides a controllable method for time series generation based on an interpretable trend-seasonality-residual decomposition of latent embeddings. We explore the interpretable decomposition of iHT on analytically generated time series datasets that have trend (T), seasonal (S) and noise (R) components, or a combination of two of them. The synthetic datasets are generated by uniformly sampling trend values, frequencies and levels of noise. We compute the decomposition error as the MSE between the output of each block of iHT and the corresponding TSR analytic component. In table 6, we compare against the traditional STL method, which requires the period of seasonality as an additional parameter. In *STL (exact)*, we compute STL with the exact period of the analytic signal. In *STL (approx)*, we provide STL with an approximate period estimated by analyzing the Fourier spectrum of the signal, a more realistic setting for time series decomposition. Our method shows the lowest trend error, with a small standard deviation with respect to *STL (approx)*, and shows comparable results with *STL (exact)*. Additionally, our method shows similar performance on the seasonality component when there is seasonality present in the dataset with respect to *STL (approx)*, and shows much better agreement when there is no seasonality present (T+R dataset).

In Figure 5 (a) and (c), we show the distribution of the trend, and residuals outputs from iHT against the ground truth components for the T+S+R dataset. In the case of the seasonality component (b), we plot the

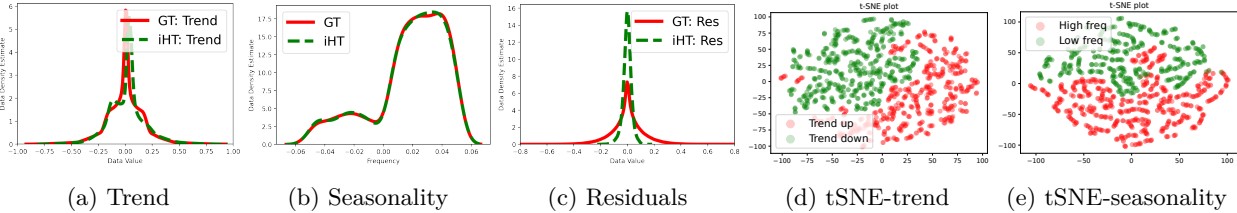

| (a) Trend | (b) Seasonality | (c) Residuals | (d) tSNE-trend | (e) tSNE-seasonality |

Figure 5: (a,b,c): Distribution of the trend, seasonality and residual outputs of iHT vs ground truth. (d,e): t-SNE visualization of trend and seasonality embeddings in iHT. All cases correspond to the T+S+R dataset.

Table 6: Ablation study for trend-seasonality-residual decomposition of time series. We compare iHT against STL decomposition, for three dataset configurations: trend+seasonality (T+S), trend+residual (T+R) and all three components (T+S+R).

| MSE ($\times 10^{-3}$) | T+S+R | | | T+R | | | T+S | | |
|---|---|---|---|---|---|---|---|---|---|
| | STL (exact) | STL (approx) | **iHT (Ours)** | STL (exact) | STL (approx) | **iHT (Ours)** | STL (exact) | STL (approx) | **iHT (Ours)** |
| Trend | $0.46 \pm 2.08$ | $1.04 \pm 4.69$ | $0.60 \pm 0.50$ | $0.07 \pm 0.07$ | $1.26 \pm 5.70$ | $0.20 \pm 0.27$ | $0.46 \pm 2.10$ | $1.17 \pm 5.12$ | $0.46 \pm 0.48$ |
| Seasonality | $0.57 \pm 2.10$ | $1.19 \pm 4.73$ | $1.21 \pm 0.68$ | $0.03 \pm 0.02$ | $1.35 \pm 5.74$ | $0.17 \pm 0.29$ | $0.44 \pm 2.09$ | $1.18 \pm 5.11$ | $1.25 \pm 0.77$ |
| Residuals | $0.17 \pm 0.15$ | $0.18 \pm 0.12$ | $1.29 \pm 0.62$ | $0.15 \pm 0.09$ | $0.18 \pm 0.13$ | $0.41 \pm 0.26$ | $0.03 \pm 0.10$ | $0.04 \pm 0.07$ | $1.25 \pm 0.71$ |

histogram for the *frequencies* of the time series, where we estimate the frequency of each time series by finding the dominant frequency component in the discrete Fourier transform. The trend and seasonality histograms show very good agreement with the ground truth, while the residuals show slightly wider tails. Finally, we visualize the learned representations via t-SNE of the embeddings. Figure 5 (d) and (e) shows that iHT is able to learn the trend and seasonal patterns from the dataset. In plot (d) the color separation corresponds to positive and negative trend, while plot (e) shows the separation in frequencies. The effectiveness of this decomposition is further showcased in our additional experiments on synthetic and real-world datasets (Appendices H and I), where each hypernet accurately captures its corresponding component. Additional evidence is provided in Appendix J through controlled generation experiments demonstrating the ability of the trend-hypernet to manipulate the trend of generated time series.

### 4.7 Ablation studies

In Table 7, we change the architecture of iHT to create simpler ablation models, and report predictive and discriminative metrics for multiple datasets. **iHT** corresponds to our full proposed model. In **iHT (no FFT)** we have removed the FFT loss from the training. In **iHT-SIREN** we remove the TSR decomposition from iHT, replacing TSNet with a SIREN network. We observe that the predictive scores are comparable for all configurations, while discriminative scores improve for all datasets when we incorporate the interpretable decomposition. The error reduces further when we add the FFT loss to the training process. The strong predictive scores across all variants indicate their effectiveness in capturing the underlying patterns and dynamics of the time series data. This suggests that even models with lower discriminative scores, such as iHT (no FFT) and iHT-SIREN, are capable of generating useful predictions. However, the discriminative scores for these variants reveal that they may struggle to produce samples that are statistically indistinguishable from real data. In particular, iHT-SIREN shows the worst discriminative score across all datasets, showcasing the importance of TSNet, while the improvement in discriminative score from the Fourier loss is more modest. We show additional ablation studies for irregularly sampled data in Section K.1 in the Appendix.

In Table 8 we study the effect of $w_0$ in iHT, with $w_0 \in [5, 10, 30, 100, 300]$, and we report the generation metrics for the Stock24 and Stock72 datasets. In Stock24, the best results are obtained with $w_0 = 30$. In the case of Stock72, the discriminative and predictive scores show a slight improvement for $w_0 = 100$ with $w_0 = 30$ showing the second best performance.

Finally, we conduct an ablation study on iHT using Random Fourier Features (FFN) instead of a SIREN network for the residuals. Given the importance of the scale factor in FFN as explained in (Woo et al., 2023), we evaluate FFN with multiple scale factor values: 5, 10, 100. Table 9 shows the discriminative and predictive

Table 7: Ablation study for model architecture: comparison of iHT against simpler configurations: *iHT (no FFT)* without FFT loss, and *iHT-SIREN* without TSR decomposition.

| | Energy24 | Stock24 | Stock72 | Stock360 | | Energy24 | Stock24 | Stock72 | Stock360 |
|---|---|---|---|---|---|---|---|---|---|
| *Predictive Score* | | | | | *Disc. Score* | | | | |
| **iHT** | 0.047 | 0.013 | 0.188 | 0.168 | **iHT** | 0.245 | 0.044 | 0.014 | 0.009 |
| **iHT (no FFT)** | 0.046 | 0.014 | 0.188 | 0.168 | **iHT (no FFT)** | 0.278 | 0.073 | 0.015 | 0.011 |
| **iHT-SIREN** | 0.048 | 0.013 | 0.188 | 0.169 | **iHT-SIREN** | 0.341 | 0.108 | 0.022 | 0.024 |

Table 8: iHT with different values of $w_0 \in [5, 10, 30, 100, 300]$ for Stock24 and Stock 72 datasets.

| | $w_0 = 5$ | $w_0 = 10$ | $w_0 = 30$ | $w_0 = 100$ | $w_0 = 300$ |
|---|---|---|---|---|---|
| *Stock24* | | | | | |
| Discr-score | 0.151±0.059 | 0.212±0.044 | 0.054±0.028 | 0.087±0.066 | 0.489±0.007 |
| Pred-score | 0.041±0.001 | 0.040±0.001 | 0.037±0.000 | 0.037±0.000 | 0.059±0.002 |
| *Stock72* | | | | | |
| Discr-score | 0.045±0.034 | 0.058±0.038 | 0.028±0.027 | 0.026±0.011 | 0.250±0.037 |
| Pred-score | 0.188±0.002 | 0.186±0.001 | 0.185±0.001 | 0.183±0.001 | 0.196±0.000 |

scores for the Stock24 and Stock72 datasets. For the Stock24 dataset, SIREN achieves the best discriminative score. In the Stock72 dataset, FFN with scale factor 10 shows slightly better performance in discriminative score than SIREN network. In all cases the predictive score shows similar performance.

Table 9: iHT with SIREN layers and with FFN with different scales ($\sigma \in [5, 10, 100]$) for Stock24 and Stock72 datasets.

| | SIREN ($w_0 = 30$) | FFN ($\sigma = 5$) | FFN ($\sigma = 10$) | FFN ($\sigma = 100$) |
|---|---|---|---|---|
| *Stock24* | | | | |
| Discr-score | 0.054±0.028 | 0.244±0.162 | 0.346±0.169 | 0.36±0.183 |
| Pred-score | 0.037±0.000 | 0.036±0.000 | 0.036±0.000 | 0.036±0.000 |
| *Stock72* | | | | |
| Discr-score | 0.028±0.027 | 0.028±0.014 | 0.024±0.026 | 0.054±0.029 |
| Pred-score | 0.185±0.001 | 0.184±0.001 | 0.185±0.002 | 0.184±0.001 |

## 5 Conclusions and Limitations

In this work, we presented iHyperTime, a versatile and efficient framework for generating time series with a wide range of characteristics. Unlike existing generative models that excel either in short or long sequences, our model demonstrates superior performance across both types of datasets. Our evaluations reveal its efficacy in handling irregularly sampled data, where it consistently surpasses current benchmarks. For regularly sampled sequences, iHyperTime's performance is competitive with the best available models, regardless of time series length. One of the model's notable strengths is its rapid training speed, which is not only comparable to the quickest existing methods for short sequences but also significantly faster for longer ones. Importantly, our architecture incorporates inductive biases that facilitate unsupervised decomposition of time series into trend, seasonality, and residual components, a capability we validated against the established STL decomposition method. Additionally, we illustrated iHyperTime's ability to learn semantically meaningful representations, opening the door for applications that involve generating time series conditioned on interpretable factors.

**Limitations and Future Work**   Despite its strengths, iHT exhibits limitations, particularly its reliance on conditional generation mechanisms, which may restrict diversity in the generated sequences. This limitation becomes particularly evident when two closely similar time series are used for generation, especially with a small interpolation parameter ($\lambda$), potentially resulting in a narrower range of diversity than desired. Such a scenario underscores the need for a more robust strategy to ensure a wider exploration of the embedding space without compromising the integrity and relevance of the generated data.

To address this limitation and enhance iHT's generative capabilities, a possible approach is to incorporate a convex combination of multiple embeddings into the framework. This approach, akin to strategies employed for data augmentation with PCA (Serrano et al., 2016), promises to expand the diversity of generated time series. By adopting a method that allows for random sampling within the convex hull of the embedding space, iHT could achieve a more comprehensive exploration of potential time series characteristics while maintaining consistency with the underlying data distribution. Moreover, inspired by competitive results obtained by diffusion models in generation tasks, integrating diffusion model principles into iHT could pave the way for the development of an advanced generative model capable of unconditional generation. This extension would not only broaden the scope of iHT's applicability but also contribute to the evolution of time series generation methodologies, offering new opportunities for research and application in various domains. Our research demonstrates promising results in time series generation across a wide range of datasets and highlights the potential of using implicit neural representations (INR) for time series generation with interpretable components. We hope our work could inspire and foster novel research in this area, which has received limited attention.

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

# A   Additional Related Work

**Implicit Neural Representations**   INRs (or coordinate-based neural networks) have recently gained popularity in computer vision applications. The usual implementation of INRs consists of a fully-connected neural network (MLP) that maps coordinates (*e.g.* xyz-coordinates) to the corresponding values of the data, essentially encoding their functional relationship in the network. One of the main advantages of this approach for data representation, is that the information is encoded in a continuous/grid-free representation, that provides a built-in non-linear interpolation of the data. This avoids the usual artifacts that arise from discretization, and has been shown to combine flexible and accurate data representation with high memory efficiency (Sitzmann et al., 2020b; Tancik et al., 2020). Whilst INRs have been shown to work on data from diverse sources, such as video, images and audio (Sitzmann et al., 2020b; Chen et al., 2021; Rott Shaham et al., 2021), their recent popularity has been motivated by multiple applications in the representation of 3D scene data, such as 3D geometry (Park et al., 2019; Mescheder et al., 2019; Sitzmann et al., 2020a; 2019) and object appearance (Mildenhall et al., 2020; Sztrajman et al., 2021). In early architectures, INRs showed a lack of accuracy in the encoding of high-frequency details of signals. Mildenhall et al. (2020) proposed positional encodings to address this issue, and Tancik et al. (2020) further explored them, showing that by using Fourier-based features in the input layer, the network is able to learn the full spectrum of frequencies from data. Concurrently, Sitzmann et al. (2020b) tackled the encoding of high-frequency data by proposing the use of sinusoidal activation functions (SIREN: Sinusoidal Representation Networks), and Benbarka et al. (2022) showed the equivalence between Fourier features and single-layer SIRENs. Our INR architecture for time series data (Section 3) is based on the SIREN architecture by Sitzmann *et al.*

**Hypernetworks**   A hypernetwork is a neural network architecture designed to predict the weight values of a secondary neural network, denominated a hyponetwork (Sitzmann et al., 2020a). The concept of hypernetwork was formalized by Ha et al. (2017), drawing inspiration from methods in evolutionary computing (Stanley et al., 2009). Moreover, while convolutional encoders have been likened to the function of the human visual system (Skorokhodov et al., 2021), the analogy cannot be extended to convolutional decoders, and some researchers have argued that hypernetworks much more closely match the behavior of the prefrontal cortex (Russin et al., 2020). Hypernetworks have been praised for their expressivity, compression due to weight sharing, and for their fast inference times(Skorokhodov et al., 2021). They have been leveraged for multiple applications, including few-shot learning (Rusu et al., 2019; Zhao et al., 2020), continual learning (von Oswald et al., 2020) and architecture search (Zhang et al., 2019; Brock et al., 2018). Moreover, in the last two years some works have started to leverage hypernetworks for the training of INRs, enabling the learning of latent encodings of data, while also maintaining the flexible and accurate reconstruction of signals provided by INRs. This approach has been implemented with different hypernetwork architectures, to learn priors over image data (Sitzmann et al., 2020b; Skorokhodov et al., 2021), 3D scene geometry (Littwin & Wolf, 2019; Sitzmann et al., 2019; 2020a) and material appearance (Sztrajman et al., 2021). Tancik et al. (2021) leverage hypernetworks to speed-up the training of INRs by providing learned initializations of the network weights. Sitzmann et al. (2020b) combine a set encoder with a hypernetwork decoder to learn a prior over INRs representing image data, and apply it for image in-painting. Our hypernetwork architecture from Section 3 is similar to Sitzmann *et al.*'s, however we learn a prior over the space of time series and leverage it for new data synthesis through interpolation of the learned embeddings.

**Interpretable Time Series**   Seasonal-trend decomposition techniques are standard tools in time series analysis used to decompose a time series into trend, seasonal, and remainder components. The trend component encapsulates the slow time-varying behavior of the time series, while seasonal components capture recurring (i.e., periodic) fluctuations in the data. These techniques enable an intuitive and interpretable analysis of time series data which play an important role in a variety of downstream tasks, including forecasting and anomaly detection. The classic approach for performing the decomposition is the widely used STL algorithm (Cleveland et al., 1990). To account for outliers and distributional shifts, a robust version of the algorithm, called Robust STL, has also been proposed (Wen et al., 2019). Additional challenges in seasonal-trend decomposition involve dealing with complex time series data that exhibit multiple seasonal components, to which techniques such as multiple STL (MSTL) have been proposed (Bandara et al., 2022). The ability to break time series into interpretable components has been a topic of recent interest in the

context of anomaly detection, forecasting, and generation. Relevant to this work is the recently proposed N-BEATS architecture (Oreshkin et al., 2020), a deep learning-based univariate time series forecasting solution that provides time series interpretability capabilities without considerable loss in predictive performance. The N-BEATS architecture explicitly encodes seasonal-trend decomposition into the network by defining two blocks: a trend block which uses a small ordered polynomial to capture slow varying behaviors, and a seasonality block which uses a Fourier series to capture cyclical patterns. Little work, however, has been done in the design of *generation* schemes that allow for decomposition of time series data into interpretable components. While TimeVAE (Desai et al., 2021) proposes a VAE architecture where the decoder has trend and seasonality blocks to allow for interpretable generation, no results highlighting the advantage of this capability were demonstrated.

**Time Series Imputation**    Time series imputation (or interpolation), which aims to fill in missing values or estimate values at specific time points within an existing series (Cao et al., 2018; Liu, 2018; Luo et al., 2018; Shukla & Marlin, 2021), is a related but distinct area of research. While both time series imputation and generation involve producing new data, they differ significantly in their goals and evaluation metrics. Time series generation focuses on creating entirely new sequences that replicate the patterns and characteristics of the original dataset, often assessed using discriminative and predictive scores. In contrast, time series imputation focuses on accurately filling gaps within existing series, typically evaluated using metrics like mean squared error. Furthermore, the methodological approaches for imputation and generation are not always directly transferable. A model designed for effective generation may not excel at imputation, and vice-versa. In SectionF, we have adapted our time series generation approach to also handle the task of imputation, demonstrating its flexibility and broad applicability.

## B    Datasets

Table 10: Main characteristics of the datasets used.

| Dataset | Number of Samples | Length of Time series | No of Features | Source |
|---|---|---|---|---|
| Stock24 | 3661 | 24 | 6 | |
| Stock72 | 3613 | 72 | 1 | Link |
| Stock360 | 3325 | 360 | 1 | |
| Energy | 19635 | 24 | 28 | Link |
| FRED-MD | 107 | 728 | 1 | Link |
| NN5 Daily | 111 | 791 | 1 | Link |
| Temp Rain | 32072 | 725 | 1 | Link |
| Solar Weekly | 137 | 52 | 1 | Link |
| PhysioNet | 8000 | $\leq 2880$ | 41 | Link,Link |
| USHCN | 1218 | $\leq 1491$ | 5 | Link |
| Atmospheric CO2 | 288 | 60 | 1 | Link |
| Airline Passenger | 108 | 36 | 1 | Link |

Here we introduce in detail the datasets used in our evaluation. We use publicly available Google stocks data from Yahoo finance, and the UCI Energy dataset (Candanedo et al., 2017). Google stock dataset contains daily observations from 2004 to 2019 with 6 features, namely *open, high, low, close, adjusted close, and volume*. The energy data contains 28 features with 10-minute resolution. Finally, we consider 4 datasets with longer real-world time-series from Monash repository (Godahewa et al., 2021), namely FRED-MD, NN5 Daily, Temperature Rain, and Solar Weekly. The first three datasets have time-series of length of around 700, while the latter one has time-series with length of 52. The datasets characteristics are summarized in Table 10.

In order to make a fair comparison with current state-of-the-art methods, we process the Stock and Energy in two different ways: in line with Yoon et al. (2019) and Jeon et al. (2022), we slice the data using a window of 24 time steps, corresponding to datasets Stock24 and Energy24. Following Coletta et al. (2023), we select one feature per dataset (univariate) and slice it using windows of 72 and 360 time steps, which correspond to Stock72 and Energy360.

In Section F we extend the evaluation of our model on two real-world irregularly sampled datasets: PhysioNet, which corresponds to health measurements from ICU patients, and USHCN, a climatology network dataset reflecting environmental conditions such as precipitation and temperature. In Section I we evaluate the decomposition capability of iHT on real-world datasets that exhibit strong trend and seasonality: Atmospheric CO2 and Airline Passanger datasets. The Atmospheric CO2 dataset corresponds to monthly data (January 1959 to December 1987) with a total of 348 observations with a seasonality period of 12 (Cleveland et al., 1990). We sliced the data into time series of sequence length 60, obtaining a total of 288 time series. The Airline Passenger dataset corresponds to monthly data (January 1948 to December 1960) with a total of 144 observations with a seasonality period of 12. We sliced the data into time series of sequence length 36, obtaining a total of 108 time series.

## C  Fourier-based loss

As part of the training of our iHyperTime architecture, we propose a Fourier spectrum reconstruction loss. For a discrete-time signal $\mathbf{f} = \{f_0 = f(0), f_1 = f(1), \ldots, f_N = f(N)\}$, the $N$-point discrete Fourier transform (DFT) is utilized to obtain the corresponding frequency domain representation of $\mathbf{f}$ through the following operation:

$$F_k = [\mathcal{F}_T\{\mathbf{f}\}]_k = \sum_{n=0}^{N-1} f_n e^{-2\pi i \left(\frac{kn}{N}\right)}, \quad 0 \leq k \leq N-1,$$

where $i = \sqrt{-1}$ corresponds to the imaginary unit of a complex number. The coefficient $F_k \in \mathbb{C}$ quantifies the strength in representation of the $k$th frequency component of the signal. The DFT has a time complexity of $\mathcal{O}(N^2)$. In practice, an algorithm called the fast Fourier transform (FFT) is used to compute the DFT due to its lower time complexity (i.e., $\mathcal{O}(N \log N)$). Using the FFT to obtain the frequency domain representations of two discrete-time signals $\mathbf{f}$ and $\hat{\mathbf{f}}$, we introduce a Fourier-based reconstruction loss as follows:

$$\mathcal{L}_{\text{FFT}} = \frac{1}{N} \sum_{k=0}^{N-1} \|F_k - \hat{F}_k\|.$$

Here, we utilized the PyTorch implementation of the FFT to obtain the DFT for each signal. It is important to note that the DFT is only well-defined for regularly sampled signals. In the case of this work, the discrete-time signal $\mathbf{f}$ is obtained by deterministically sampling the function $f(t)$ via a discretized grid of time steps $t \in \{0, 1, \ldots, N\}$.

## D  Implementation & Reproducibility Details

### D.1  Baselines

We use the following methods with publicly available code as benchmark for our method:

- Fourier Flows (Alaa et al., 2021): `https://github.com/ahmedmalaa/Fourier-flows`

- TimeGAN (Yoon et al., 2019): `https://github.com/jsyoon0823/TimeGAN`

- GT-GAN(Jeon et al., 2022): `https://github.com/Jinsung-Jeon/GT-GAN`

- RCGAN (Esteban et al., 2017): `https://github.com/3778/Ward2ICU`

- LS4(Zhou et al., 2023): `https://github.com/alexzhou907/ls4/tree/main`

- DiffTime(Coletta et al., 2023): `https://arxiv.org/abs/2307.01717`

We adapted *TimeGAN*, *RCGAN*, and *GT-GAN* for longer time-series by setting the hidden dimensions to be equal to the time-series length, as suggested by authors and empirically evaluated. Moreover, we improve RCGAN discriminator to handle longer time-series more effectively using the CSDI transformer architecture (Tashiro et al., 2021). For DiffTime we reach out the authors to get access to their code, and to handle missing data we dynamically mask the input time-series to let the model learn to reconstruct the whole original time-series, similarly to CSDI approach for imputation (Tashiro et al., 2021).

### D.2    Implementation Details

iHyperTime is composed of a set encoder, three decoder (hypernetworks) whose outputs corresponds to the weights of each of the blocks in TSNet. Below we explain each component in detail.

**Set Encoder**    The set encoder is a SIREN with two hidden layers of 128 neurons and an output layer (embedding) of 40 neurons. It takes as input an arbitrary set of tuples $\{(t_i, \mathbf{y}_i)\}_{i=1}^N$, where $t$ denotes the time-coordinate and $\mathbf{y}_i$ the corresponding univariate or multivariate time series value. We use a single floating point value as temporal coordinate (time $t$). As data pre-processing, the values are scaled to the interval $[-1, 1]$, with a common global factor for all time series of a dataset. MinMax scaling is also applied to the time series amplitudes, although in the interval $[0, 1]$. In all cases, regardless of sequence length, the time series is fed to the set encoder as a single set, and is hence converted into a single embedding $Z$.

**Decoder (Hypernetwork)**    Each decoder block (hypernetwork) is a one-layer MLP with ReLU activations, with a hidden layer of dimension 128. The output of each hypernetwork is a vector that contains the weights of its corresponding decomposition block. Table 11 shows the dimension details, where $n_t$, $n_S$ and $n_R$ correspond to the number of weights in the trend, seasonality and residuals blocks, which form TSNet.

Table 11: Architecture of the hypernetworks in the decoder.

| Hypernet block | Design | Input size | Output size |
|---|---|---|---|
| Trend Hypernet | Relu | 10 | 128 |
| | Linear | 128 | $n_T$ |
| Season Hypernet | Relu | 15 | 128 |
| | Linear | 128 | $n_S$ |
| Res. Hypernet | Relu | 15 | 128 |
| | Linear | 128 | $n_R$ |

**TSnet Architecture**    TSnet is an implicit neural representation of univariate/multivariate time series data. It is composed of three distinctive blocks that perform a trend-seasonality-residual additive decomposition of the time series signal. Table 12 shows the network details of each component of TSNet. In the trend block, $p$ corresponds to the degree of the polynomial, $L$ corresponds to the max length of the time series, and $m$ corresponds to the number of features.

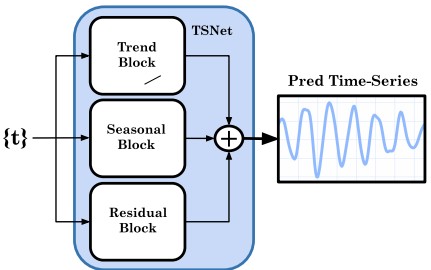

Figure 6: Diagram of the TSnet architecture.

Table 12: Architecture of TSNet.

|  | Layer | Design | Input size | Output size |
|---|---|---|---|---|
| Trend block | 1 | Linear | $p$ | $m$ |
| Seasonality block | 1 | Linear | $L$ | $m$ |
| Residual block | 1 | Sine(Linear) | 1 | 60 |
|  | 2 | Sine(Linear) | 60 | 60 |
|  | 3 | Sine(Linear) | 60 | 60 |
|  | 4 | Sine(Linear) | 60 | 60 |
|  | 5 | Linear | 60 | $m$ |

**Training** The training of iHT is performed in three stages to improve stability: 1) we train the Trend HyperNetwork, Trend Block for 100 epochs, computing the MSE loss between the ground truth time series $\mathbf{y}$ and the output of the block: $\mathcal{L}_1 = \sum_i \|y_i - \hat{f}_{\text{tr}}(t)\|^2$. This leads to a smooth approximation of the time series, which we use as initial guess for the second stage. 2) We then train the Trend and Seasonality blocks together, computing the MSE reconstruction loss $\mathcal{L}_{\text{rec}}$ and the FFT loss $\mathcal{L}_{\text{FFT}}$ between the ground truth and the added output of both TSnet blocks. 3) Finally, we train the three blocks together.

- Stage 1 training:
    - Number of epochs: 100
    - Learning rate: $1e - 3$

- Stage 2 training:
    - Number of epochs: 150
    - Learning rate: $5e - 5$

- Stage 3 training:
    - Number of epochs: 150
    - Learning rate: $5e - 5$

- Batch size: 256

- $\lambda_1 = 1.0 \times 10^{-3}$

- $\lambda_2 = 1.0$

- $\lambda_3 = 1.0 \times 10^{-2}$

We train Energy24, Stock24, Stock72, Stock360 and Solar Weekly datasets for 400 epochs, with Adam optimizer. For the NN5 daily, and Fred MD datasets we trained for 500 epochs, and for the Temperature Rain dataset we train iHT for 1500 epochs.

**Hardware and Software** We implement our method in Python and the experiments are ran using a *g4dn.2xlarge* AWS instance with a NVIDIA T4 GPU, 8 CPU and 32gb of RAM.

## E Additional experimental results on real-world time series synthesis

We show additional comparisons of time series synthesis the four Monash datasets in Table 13. Out method still shows competitive results across most datasets, with best predictive score in three cases, only loosing against Latent ODE in the FRED-MD dataset.

Table 13: Generation results on FRED-MD, NN5 Daily, Temperature Rain, and Solar Weekly.

| Data | Metric | RNN-VAE | GP-VAE | ODE²VAE | Latent ODE | TimeGAN | SDEGAN | SaShiMi | LS4 | **iHT (Ours)** |
|---|---|---|---|---|---|---|---|---|---|---|
| FRED-MD | Marginal ↓ | 0.132 | 0.152 | 0.122 | 0.0416 | 0.0813 | 0.0841 | 0.0482 | 0.0221 | **0.0177** |
| | Class. ↑ | 0.0362 | 0.0158 | 0.0282 | 0.327 | 0.0294 | 0.501 | 0.00119 | 0.544 | **1.3278** |
| | Prediction ↓ | 1.47 | 2.05 | 0.567 | **0.0132** | 0.0575 | 0.677 | 0.232 | 0.0373 | 0.0181 |
| NN5 Daily | Marginal ↓ | 0.137 | 0.117 | 0.211 | 0.107 | 0.0396 | 0.0852 | 0.0199 | **0.00671** | 0.00893 |
| | Class. ↑ | 0.000339 | 0.00246 | 0.00102 | 0.000381 | 0.00160 | 0.0852 | 0.0446 | **0.636** | 0.4982 |
| | Prediction ↓ | 0.967 | 1.169 | 1.19 | 1.04 | 1.34 | 1.01 | 0.849 | 0.241 | **0.2349** |
| Temp Rain | Marginal ↓ | 0.0174 | 0.183 | 1.831 | **0.0106** | 0.498 | 0.990 | 0.758 | 0.0834 | 0.2978 |
| | Class. ↑ | 0.00000212 | 0.0000123 | 0.0000319 | 0.0000419 | 0.00271 | 0.0169 | 0.0000167 | 0.976 | **11.2493** |
| | Prediction ↓ | 159 | 2.305 | 1.133 | 145 | 1.96 | 2.46 | 2.12 | 0.521 | **0.132** |
| Solar Weekly | Marginal ↓ | 0.0903 | 0.308 | 0.153 | 0.0853 | 0.0496 | 0.147 | 0.173 | 0.0459 | **0.03273** |
| | Class. ↑ | 0.0524 | 0.000731 | 0.0998 | 0.0521 | 0.6489 | 0.591 | 0.00102 | 0.683 | **1.2413** |
| | Prediction ↓ | 1.25 | 1.47 | 0.761 | 0.973 | 0.237 | 0.976 | 0.578 | 0.141 | **0.0739** |

# F   Additional experiments with real irregularly sampled data

In this section, we present additional results on real-world datasets that are inherently irregularly sampled, such as PhysioNet and USHCN. Given that the established evaluation of time series generation (e.g., through discriminative and predictive scores) is not possible in these datasets, we focus on the task of interpolation, following the approach taken in prior works (e.g., De Brouwer et al. 2019, Zhou et al. 2024, Rubanova et al. 2019). In particular, we adopt the experimental setup outlined in Zhou et al. (2023).

**Data**   Following (Zhou et al., 2023), we use PhysioNet and USHCN as our datasets of choice. The United States Historical Climatology Network (USHCN) (Menne et al., 2016) is a climate dataset containing daily measurements from 1,218 weather stations across the US for precipitation, snowfall, snow depth, minimum, and maximum temperature. PhysioNet (Silva et al., 2012) is a dataset containing health measurements of 41 signals from 8,000 ICU patients in their first 48 hours. We follow the code provided by (Zhou et al., 2023)[1] to process PhysioNet and the code provided by (Brouwer et al., 2019)[2] to pre-process USHCN. We use mean squared error (MSE) to evaluate.

Table 14: MSE scores $\times 10^{-3}$ for interpolation on irregularly sampled time series data.

| Data | RNN | RNN-VAE | ODE-RNN | GRU-D | Latent ODE | LS4 | **iHT (Ours)** |
|---|---|---|---|---|---|---|---|
| PhysioNet | 2.92 | 5.93 | 2.23 | 3.33 | 8.34 | 0.63 | 3.35 |
| USHCN | 4.32 | 7.56 | 2.47 | 3.40 | 6.86 | 0.06 | 1.46 |

We compare iHT with RNN (Rumelhart et al., 1986), RNN-VAE (Chung et al., 2014; Rubanova et al., 2019), ODE-RNN (Rubanova et al., 2019), GRU-D (Rubanova et al., 2019), Latent ODE (Chen et al., 2018; Rubanova et al., 2019) and LS4. Table 14 presents the interpolation performance of various models on irregularly sampled time series from the PhysioNet and USHCN datasets. While LS4 consistently achieves the best performance, our proposed model demonstrates strong competitiveness. On PhysioNet, iHT's MSE of $3.35 \times 10^{-3}$ outperforms several advanced models, including Latent ODE and RNN-VAE. On USHCN, iHT achieves the second-best performance with an MSE of $1.46 \times 10^{-3}$. These results point towards iHT's capability to effectively handle irregular sampling across diverse domains, further demonstrating its potential for real-world applications.

# G   Training time comparison

Table 15 shows the training time of iHT and all the other baselines for the Energy and Stock datasets. iHT has the lowest training time across all datasets, with Fourier Flows having a similar performance in the Stock

---

[1] https://github.com/alexzhou907/ls4/blob/main/README.md
[2] https://github.com/edebrouwer/gru_ode_bayes

datasets. In the case of Energy, given that Fourier Flows trains on each feature separately, this sequential training increases the computational time because of the large number of features present in the dataset. The training times of TimeGAN and GTGAN are orders of magnitude larger for the datasets with the longest time series, in the case of TimeGAN because it based on RNNs, whilst GTGAN's needs to solve various differential equations.

Table 15: Comparison of training time. iHT shows the shortest training time on all datasets.

| Training Time (HH:MM) | Energy24 | Stock24 | Stock72 | Stock360 |
|---|---|---|---|---|
| **iHT (Ours)** | **00:15** | **00:03** | **00:03** | **00:04** |
| GTGAN | 10:39 | 12:20 | 04:32 | 21:23 |
| TimeGAN | 12:28 | 11:40 | 34:30 | 65:00 |
| FourierFlows | 02:48 | 00:07 | **00:03** | 00:05 |
| LS4 | 04:19 | 00:57 | 01:16 | 02:09 |
| DiffTime | 17:03 | 02:52 | 01:42 | 02:13 |

## G.1 Additional Analysis of Computational Times

Our experiments shown in Figure4 use the same synthetic data-set of 8192000 data-points (e.g. number of samples $\times$ sequence length), organized in 5 datasets of different sequence lengths $l \in \{80, 320, 1280, 5120, 20480\}$. Thus, we obtain 5 datasets of $\{102400, 25600, 6400, 1600, 400\}$ number of samples, which we process using $\{1024, 256, 64, 16, 4\}$ as batch-size, respectively. Thanks to this batch-size, we can fairly train each model for 100 iteration and always loading to GPU the same amount of data-points, for each sequence length and iteration.For inference, we only sample a single batch of data.

**Impact of data loading** All the times are computed directly in the training and testing loop, after the model has been already loaded in the GPU memory, using a NVIDIA T4 GPU. Moreover, we found that the data loading (both for training and inference) is negligible w.r.t. model computational time. We measure the loading being around 0.0145 sec for training (100 iterations) and inference 0.001 sec (1 iteration).

**Fourier Flow** Finally, we add in Table 16 the computational times Fourier Flows, which trend still has some convexing. We believe the slightly drop likely comes from not optimized code to handle large batch-sizes: the longer time-series (e.g., 320 and 1280) worsen the computational time, but the model is also relieved by the smaller batch-size (e.g., 256 and 64).

Table 16: Left: Characteristics of datasets to assess computational time. Right: Details of training and inference times of Fourier Flows.

| Time series length | Batch size | FFlows train time (sec) | FFlows inference time (sec) |
|---|---|---|---|
| 80 | 1024 | 12.5 | 0.34 |
| 320 | 256 | 8.14 | 0.10 |
| 1280 | 64 | 6.24 | 0.08 |
| 5120 | 16 | 10.19 | 1.15 |
| 20480 | 4 | 93.72 | 39.24 |

## H  Additional Trend-Seasonality decomposition analysis

In this section we provide further details of the analysis of iHT decomposition.

**Synthetic dataset**  We generated time series datasets that have trend (T), seasonal (S) and noise (R) components, or a combination of two of them. The trend was generated by randomly choosing the degree of the polynomial, a sign and a slope. A code example with the parameters is shown in Code Snippet 1. To model the seasonality component we use a Sine function with the frequency sampled uniformly within [1,10]. Finally, for the residual component we used Gaussian noise, with the standard deviation sampled between 0 and 0.2. For each dataset, we generated 2000 time series of 200 time steps. Figure 7 shows examples of each dataset.

Code Snippet 1: Trend generation

```
1  def generate_trend():
2      trend_slope = np.random.uniform(1.5,2)
3      degree = np.random.choice([1,2,3])
4      sign = np.random.choice((-1, 1))
5      trend = sign * (trend_slope*regular_time_samples)**degree
6      return trend
```

Table 17, we compare iHT with *STL (exact)* and *STL (approx)*, in the previous dataset TSR, T+R, T+S the additional dataset of S+R dataset. We can observe that in the S+R dataset iHT shows the worst performance in all three components. Interestingly, both STL (exact) and STL (approx) show higher errors than in the other datasets, showing that this is a more challenging case.

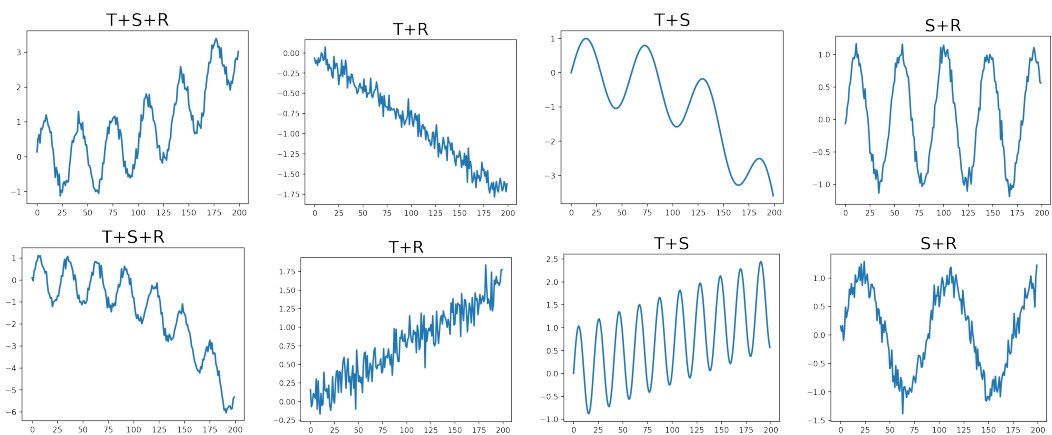

Figure 7: Example of synthetic datasets with trend (T), seasonal (S) and noise (R) components, or a combination of two of them.

Table 17: Ablation study for trend-seasonality-residual decomposition of time series. We compare iHT against STL decomposition, for three dataset configurations: trend+seasonality (T+S), trend+residual (T+R), seasonality+residual (S+R) and all three components (TSR).

| MSE ($\times 10^{-3}$) | TSR | | | T+R | | | T+S | | | S+R | | |
| --- | --- | --- | --- | --- | --- | --- | --- | --- | --- | --- | --- | --- |
| | STL (exact) | STL (approx) | **iHT (Ours)** | STL (exact) | STL (approx) | **iHT (Ours)** | STL (exact) | STL (approx) | **iHT (Ours)** | STL (exact) | STL (approx) | **iHT (Ours)** |
| Trend | $0.46 \pm 2.08$ | $1.04 \pm 4.69$ | $0.6 \pm 0.5$ | $0.07 \pm 0.07$ | $1.26 \pm 5.7$ | $0.2 \pm 0.27$ | $0.46 \pm 2.1$ | $1.17 \pm 5.12$ | $0.46 \pm 0.48$ | $0.18 \pm 0.19$ | $0.71 \pm 1.9$ | $8.11 \pm 8.31$ |
| Seasonality | $0.57 \pm 2.1$ | $1.19 \pm 4.73$ | $1.21 \pm 0.68$ | $0.03 \pm 0.02$ | $1.35 \pm 5.74$ | $0.17 \pm 0.29$ | $0.44 \pm 2.09$ | $1.18 \pm 5.11$ | $1.25 \pm 0.77$ | $2.54 \pm 2.68$ | $3.89 \pm 3.93$ | $14.86 \pm 11.49$ |
| Residuals | $0.17 \pm 0.15$ | $0.18 \pm 0.12$ | $1.29 \pm 0.62$ | $0.15 \pm 0.09$ | $0.18 \pm 0.13$ | $0.41 \pm 0.26$ | $0.03 \pm 0.1$ | $0.04 \pm 0.07$ | $1.25 \pm 0.71$ | $2.64 \pm 2.73$ | $3.45 \pm 3.07$ | $13.98 \pm 5.71$ |

In Figures 8, 9, 10 and 11 we show the evaluation of the output of iHT for each dataset. The first three plots on each row correspond to the distribution of trend, seasonality and residuals outputs from iHT against the ground truth components. We can see in Figures 9 and 10 that for the T+S and T+R datasets, the trend shows good agreement, and in the case of no residuals, we observe a narrow distribution close to zero, while

in the case of no seasonality, the frequency histogram is also very narrow around zero. In Figure 11, which corresponds to S+R we can observe that the distribution of trend is quite broad, and this is in line with the results on Table 17. Even in the seasonality distribution we can observe a slightly worse match with regards to the other cases. The plots on the right show the learned representations via t-SNE of the embeddings. We can still observe a separation in trend up and down on the T+R dataset, although the separation in high and low frequency in the S+R plot is less obvious.

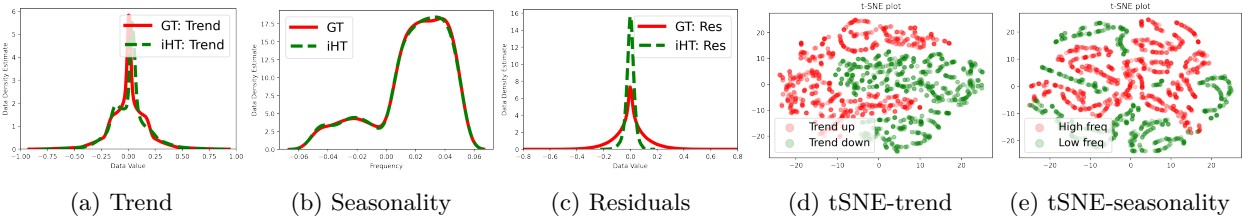

(a) Trend  (b) Seasonality  (c) Residuals  (d) tSNE-trend  (e) tSNE-seasonality

Figure 8: (a,b,c): Distribution of the trend, seasonality and residual outputs of iHT vs ground truth. (d,e): t-SNE visualization of trend and seasonality embeddings in iHT for the TSR dataset.

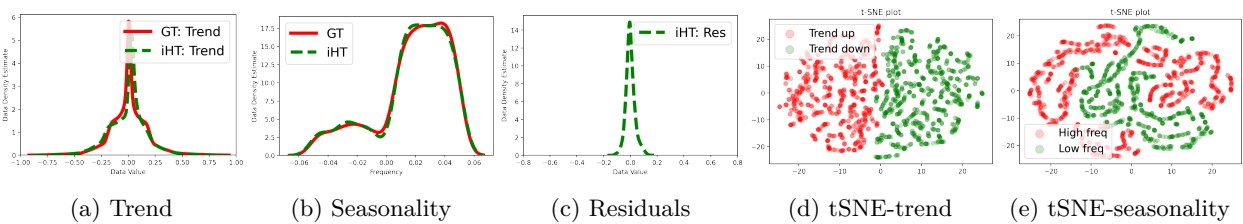

(a) Trend  (b) Seasonality  (c) Residuals  (d) tSNE-trend  (e) tSNE-seasonality

Figure 9: (a,b,c): Distribution of the trend, seasonality and residual outputs of iHT vs ground truth. (d,e): t-SNE visualization of trend and seasonality embeddings in iHT for the T+S dataset.

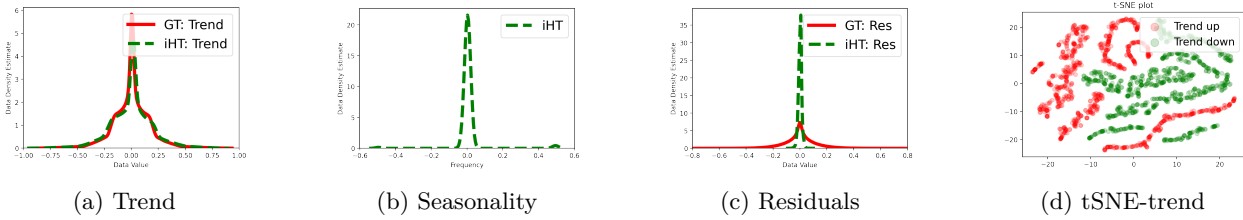

(a) Trend  (b) Seasonality  (c) Residuals  (d) tSNE-trend

Figure 10: (a,b,c): Distribution of the trend, seasonality and residual outputs of iHT vs ground truth. (d): t-SNE visualization of trend embeddings in iHT for the T+R dataset.

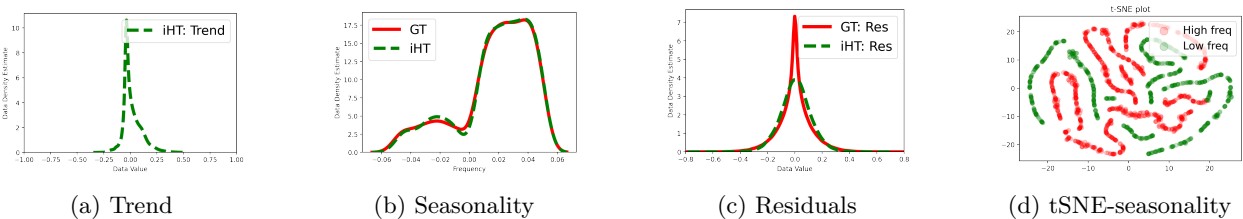

(a) Trend  (b) Seasonality  (c) Residuals  (d) tSNE-seasonality

Figure 11: (a,b,c): Distribution of the trend, seasonality and residual outputs of iHT vs ground truth. (d): t-SNE visualization of seasonality embeddings in iHT for the S+R dataset.

# I  Real-Data Trend-Seasonality decomposition analysis

In this section we report an analysis on three real world datasets to show that iHT is able to disambiguate between trend, season and residual components after training. We used two well known real-world time series datasets that exhibit strong seasonality and trend, Atmospheric CO2, and Airline Passenger, and use the Stock72 dataset that exhibits strong trend and weak seasonality, as described in section B.

The Atmospheric CO2 dataset corresponds to monthly data (January 1959 to December 1987) with a seasonality period of 12 (Cleveland et al., 1990). We sliced the data into time series of sequence length 60.

The Airline Passenger dataset corresponds to monthly data (January 1948 to December 1960) with a seasonality period of 12 [3]. We sliced the data into time series of sequence length 36.

To estimate the presence of trend and seasonality of each dataset we use two metrics: the strength of trend and strength of seasonality (Hyndman & Athanasopoulos, 2018; Wang et al., 2006). As these datasets do not include ground-truth trend and seasonality, we estimate them using STL (Cleveland et al., 1990). Given a time series with additive decomposition in trend, seasonality and residual: $y_t = T_t + S_t + R_t$ , we can define the strength of trend as:

$$F_T = \max\left(0, 1 - \frac{\text{Var}(R_t)}{\text{Var}(T_t + R_t)}\right)$$

This measures the relative variance of the remainder component $R_t$ to the variance of the trend with the remainder component. The value ranges from 0 to 1, with 0 indicating no trend and 1 indicating a strong trend.

Strength of seasonality is defined by:

$$F_S = \max\left(0, 1 - \frac{\text{Var}(R_t)}{\text{Var}(S_t + R_t)}\right)$$

Which measures the relative variance of the remainder component $R_t$ to the variance of the seasonality with the remainder component. A value close to 0 indicates little to no seasonality, and a value close to 1 indicates strong seasonality.

Table 18 shows these metrics for the three datasets. As expected, Stock72 doesn't exhibit strong seasonality, while the other two datasets exhibit strong trends and seasonality, with all values on average above 0.98.

Table 18: Strength of trend and seasonality in the CO2, Air Passanger and Stock72 datasets.

|  | CO2 | Air Pass | Stock72 |
| --- | --- | --- | --- |
| Trend strength | $0.99 \pm 0.01$ | $0.98 \pm 0.01$ | $0.99 \pm 0.01$ |
| Seasonal strength | $0.99 \pm 0.00$ | $0.99 \pm 0.01$ | $0.32 \pm 0.09$ |

We trained iHT on each dataset and we show in Figures 12 and 13 the reconstructed time series and the output of each of the individual blocks for the CO2 and Air Passanger datasets, respectively. For the output of the trend, seasonality and residual block, we compare with the STL method. We observe that there is a good agreement in the trend and seasonality components in both datasets. The STL method requires as parameter the period of the seasonality, which is known for both datasets, while our approach is able to estimate the seasonality from the data. In the case of the Stock72 dataset, for the STL comparison, we estimated the period using a Fourier transform and retrieving the dominant frequency. Figure 14 shows the decomposition generated by iHT and compared against STL. We can see in this case that the seasonal component is quite small in magnitude compared to the residuals.

---

[3]Downloaded from `https://www.kaggle.com/datasets/rakannimer/air-passengers`

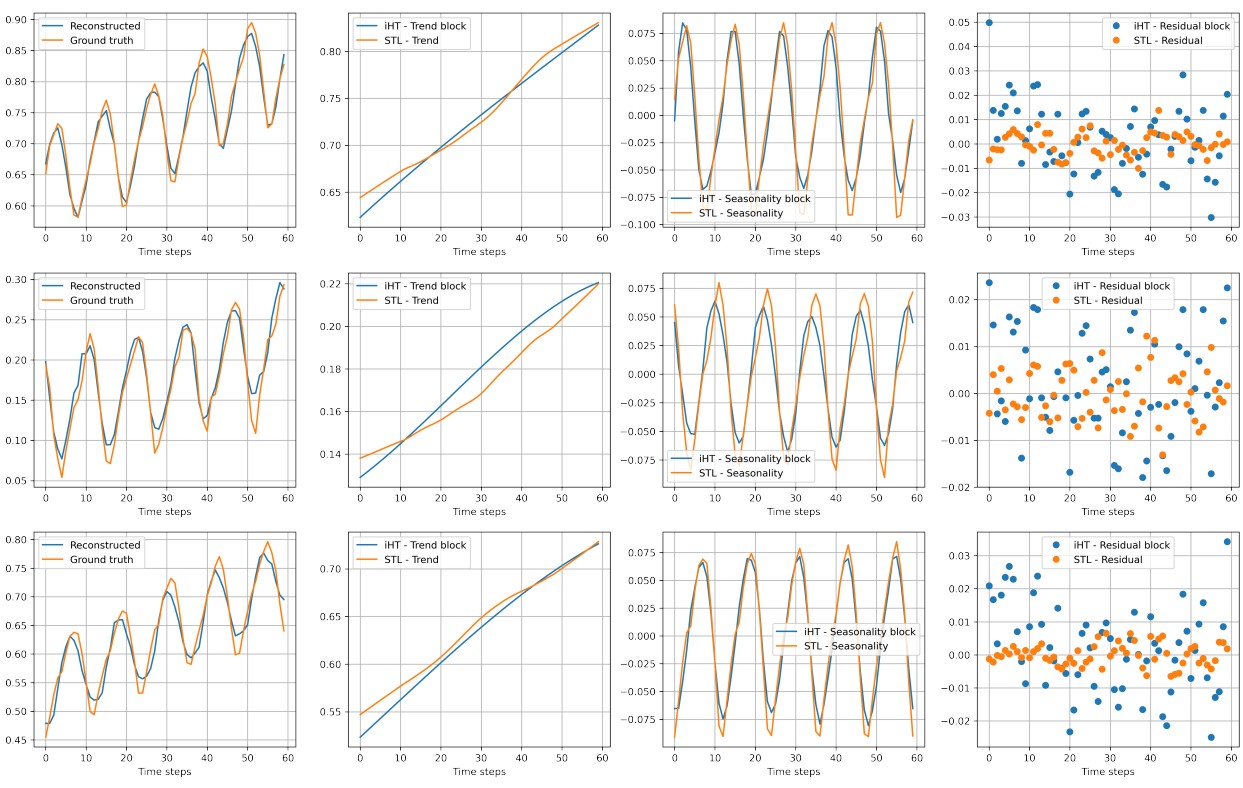

Figure 12: iHT decomposition on CO2 dataset.

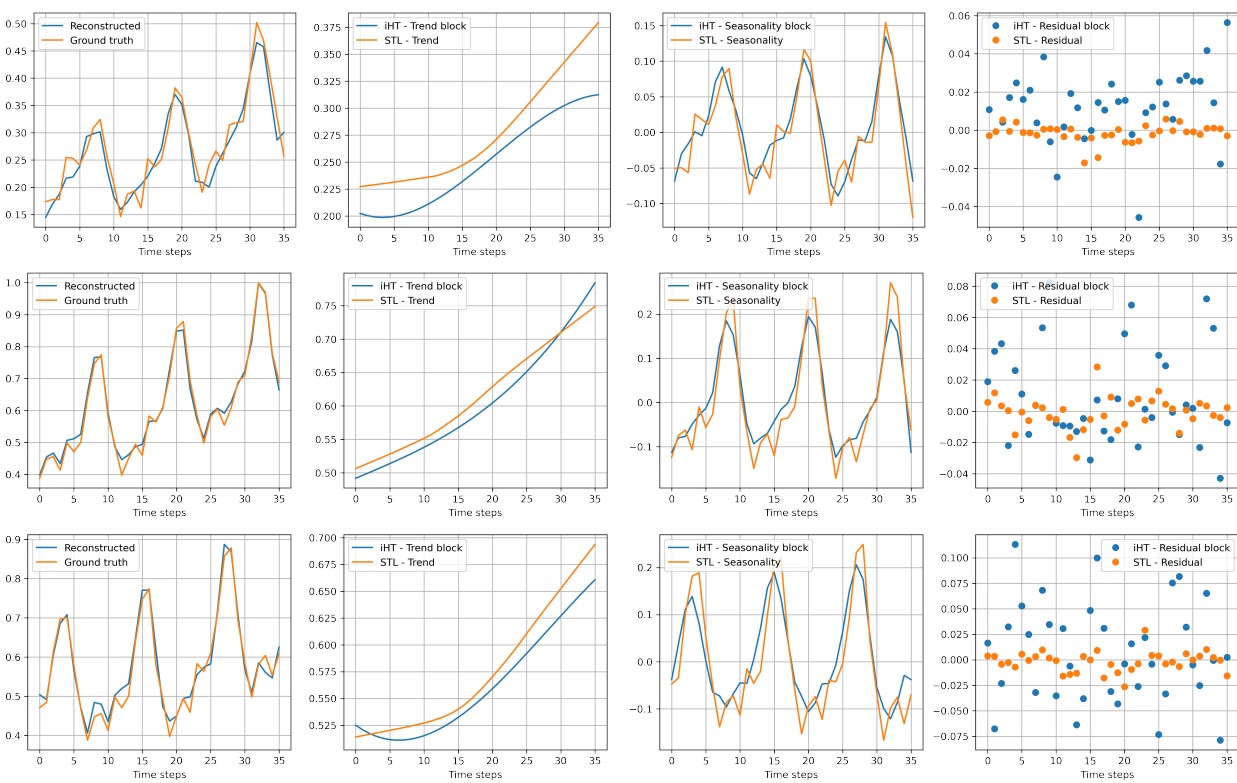

Figure 13: iHT decomposition on Air travel dataset.

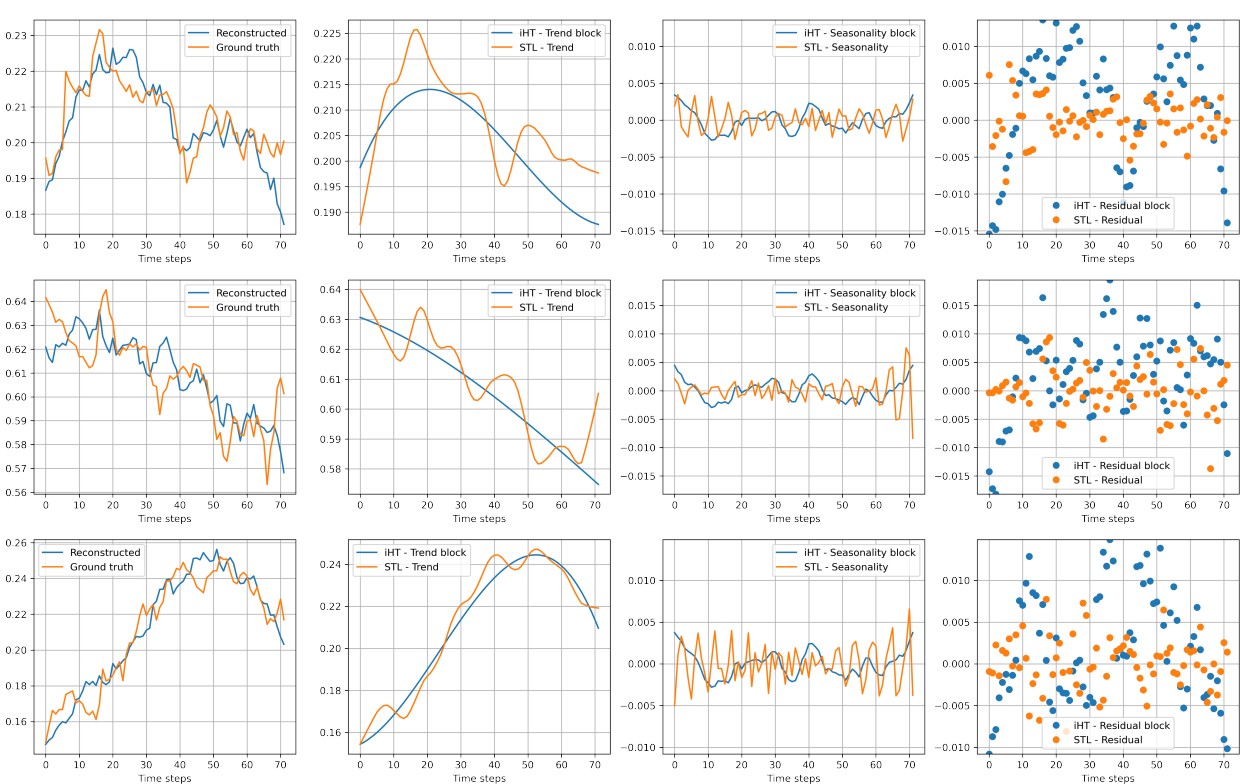

Figure 14: iHT decomposition on Stock72 dataset.

## J   Time Series Generation using Trend Component

Here we consider a peculiar capability of iHT that enables the user to provide an input trend component to generate time-series accordingly. iHT generates new time series by performing interpolation in the embedding space between two time series, and then generating the novel weights of TSNet that represents the novel time series. This allows us to control the generation by projecting a desired trend pattern in iHT to use in the generation process. To evaluate the performance of this guided generation, we use iHT trained on Stock24. We provide an input trend and generate 1000 time series using iHT. In this experiment we consider the following baselines: DiffTime, RCGAN, TimeGAN, GT-GAN. While DiffTime (Coletta et al., 2023) is naturally designed for constrained time-series generation, we adapted RCGAN, TimeGAN, GT-GAN architectures to deal with the input trend. In detail, we re-trained them as conditioned models using an additional input trend, computed as a polynomial interpolation from original data during the training. In Figure 15 we show how the generated time-series follow the trend. For each approach we generate 1000 samples and we plot their 5-95th percentile values as the light-blue shaded area, while trend is the dotted orange line. In Table 19 we report the quantitative metrics, which evaluate how much each generated time-series deviate from the input trend by computing the L2 distance and Dynamic-Time-Warping (DTW) distance between the generated sample and the input trend. The results show that iHT has among the best performance and it is competitive w.r.t. to DiffTime, which is specifically designed to incorporate such trend constraints.

Table 19: Time-Series Trend Generation on Stock24 dataset.

| Algo | L2 Distance | DTW Distance |
|------|-------------|--------------|
| iHT (Ours) | 23.68±18.6 | **14.43±13.60** |
| DiffTime | **19.83±5.40** | 15.42±4.79 |
| GT-GAN | 1304.2±1026.9 | 1303.9±1303.1 |
| TimeGAN | 88.18±12.10 | 87.29±12.35 |
| RCGAN | 60.56±9.20 | 32.94±6.05 |

Figure 15 shows additional qualitative results using iHT trained on Stock72 with a wide diversity of input trends. We can see that in all cases, the input trend is within the 5-95th percentile values of the generated time series, showing a good agreement of the synthetic time series with the input trend.

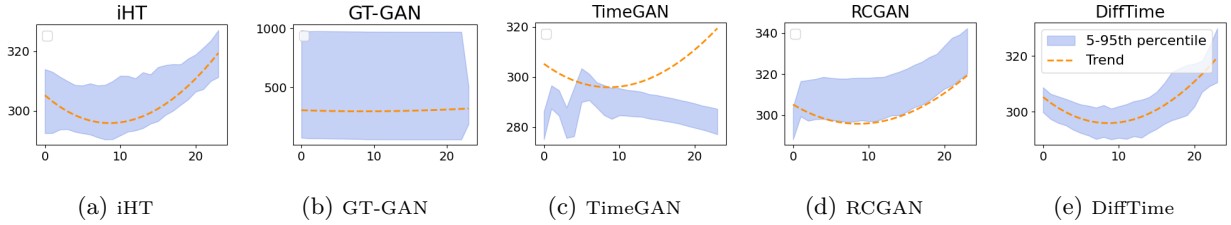

(a) iHT        (b) GT-GAN        (c) TimeGAN        (d) RCGAN        (e) DiffTime

Figure 15: A visualizations of time-series generated according the input *Trend*. The orange dotted time-series is the trend, and the shaded blue area shows the 5% and 95% percentiles of the generated synthetic time-series. Our approaches show among the best performance with time-series closer to the input trend.

## K   Additional Ablation Studies

In this section we provide further ablation studies. Firstly we report the impact in performance of the architecture of iHT on irregularly sampled data. We then study the impact in performance of the size of the latent dimension. Furthermore, we report generation metrics when splitting the dataset into train-test, including a diversity metric.

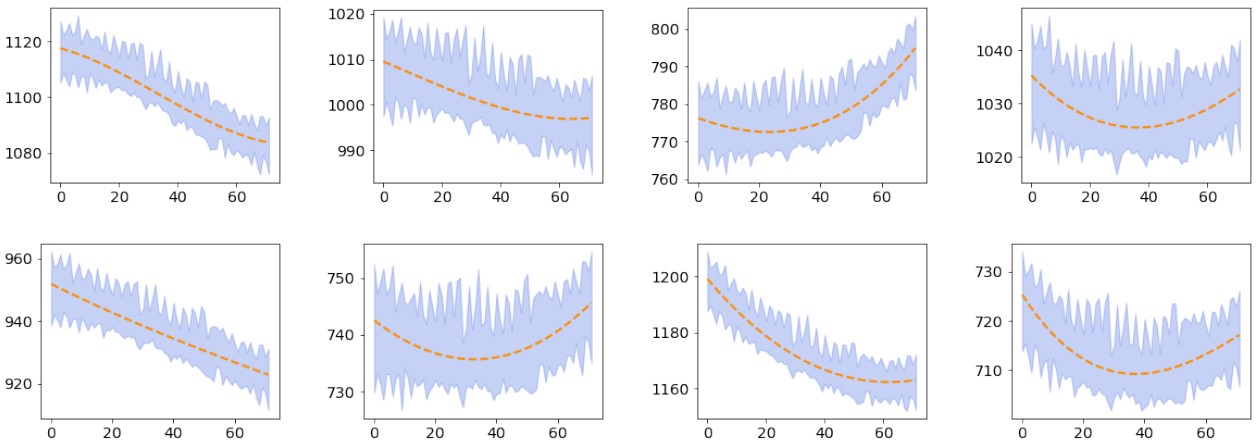

Figure 16: A visualizations of time-series generated by iHT according to an input *Trend*, on the Stock72 dataset. The orange dotted time-series is the trend, and the shaded blue area shows the 5% and 95% percentiles of the generated synthetic time-series.

### K.1 Analysis of model architecture

In Tables 20, 21 and 22 we report the predictive and discriminative scores for two simpler iHT models as explained in Section 4.7 for the case of irregularly sampled data with different levels of dropped values (30, 50 and 70, respectively). In all cases of missing values, we don't observe a difference in performance for the predictive score in the case of iHT (no FFT), but we do observe some loss in performance for the case of the simpler architecture without TSR decomposition for the Stock datasets (iHT-SIREN). In the case of the discriminative score, Energy24 dataset shows a degradation in performance for both cases of iHT-SIREN and iHT (no FFT). For Stock24 we see a degradation in performance for the iHT-SIREN model, and in the case of a small percentage of missing values there is a degradation in performance when not including the FFT loss. In the case of larger amounts of missing values, we see the opposite, that not using the Fourier loss yields higher discriminative scores. Similarly for the longer time series datasets of Stock72 and Stock360 we observe a loss in performance in discriminative score when using the simpler iHT-SIREN but in the case of iHT (no FFT) there is no clear trend.

Table 20: Ablation study for model architecture: comparison of iHT against simpler configurations: *iHT (no FFT)* without FFT loss, and *iHT-SIREN* without TSR decomposition, for irregularly sampled data with 30% dropped values.

| | Energy24 | Stock24 | Stock72 | Stock360 | | Energy24 | Stock24 | Stock72 | Stock360 |
|---|---|---|---|---|---|---|---|---|---|
| *Predictive Score* | | | | | *Disc. Score* | | | | |
| **iHT** | 0.049 | 0.013 | 0.188 | 0.168 | **iHT** | 0.452 | 0.059 | 0.017 | 0.014 |
| **iHT (no FFT)** | 0.050 | 0.014 | 0.188 | 0.167 | **iHT (no FFT)** | 0.470 | 0.068 | 0.010 | 0.007 |
| **iHT-SIREN** | 0.050 | 0.015 | 0.189 | 0.169 | **iHT-SIREN** | 0.473 | 0.089 | 0.036 | 0.008 |

Table 21: Ablation study for model architecture: comparison of iHT against simpler configurations: *iHT (no FFT)* without FFT loss, and *iHT-SIREN* without TSR decomposition, for irregularly sampled data with 50% dropped values.

| | Energy24 | Stock24 | Stock72 | Stock360 | | Energy24 | Stock24 | Stock72 | Stock360 |
|---|---|---|---|---|---|---|---|---|---|
| *Predictive Score* | | | | | *Disc. Score* | | | | |
| **iHT** | 0.051 | 0.014 | 0.187 | 0.168 | **iHT** | 0.472 | 0.102 | 0.011 | 0.004 |
| **iHT (no FFT)** | 0.051 | 0.012 | 0.188 | 0.167 | **iHT (no FFT)** | 0.482 | 0.065 | 0.019 | 0.011 |
| **iHT-SIREN** | 0.050 | 0.014 | 0.189 | 0.169 | **iHT-SIREN** | 0.481 | 0.106 | 0.017 | 0.005 |

Table 22: Ablation study for model architecture: comparison of iHT against simpler configurations: *iHT (no FFT)* without FFT loss, and *iHT-SIREN* without TSR decomposition, for irregularly sampled data with 70% dropped values.

| | Energy24 | Stock24 | Stock72 | Stock360 | | Energy24 | Stock24 | Stock72 | Stock360 |
|---|---|---|---|---|---|---|---|---|---|
| *Predictive Score* | | | | | *Disc. Score* | | | | |
| **iHT** | 0.053 | 0.014 | 0.187 | 0.168 | **iHT** | 0.482 | 0.115 | 0.020 | 0.011 |
| **iHT (no FFT)** | 0.052 | 0.015 | 0.188 | 0.168 | **iHT (no FFT)** | 0.493 | 0.062 | 0.013 | 0.008 |
| **iHT-SIREN** | 0.052 | 0.069 | 0.188 | 0.170 | **iHT-SIREN** | 0.491 | 0.479 | 0.021 | 0.017 |

## K.2 Analysis of the dimension of $Z_T$, $Z_S$ and $Z_R$

The results of varying the value of the three latent dimensions are shown in Table 23 and Table 24, where the we report $Z\_T\_S\_R$ (e.g., $Z\_4\_6\_8$ represents $Z_T = 4$, $Z_S = 6$ and $Z_R = 8$) .

Table 23: Results for different values of $Z$ on Stock 24 dataset.

| | $Z\_2\_3\_3$ | $Z\_4\_6\_6$ | $Z\_10\_15\_15$ | $Z\_40\_60\_60$ | $Z\_100\_150\_150$ |
|---|---|---|---|---|---|
| Pred score | 0.037±0.000 | 0.038±0.000 | 0.037±0.000 | 0.036±0.000 | 0.037±0.000 |
| Discr score | 0.061±0.034 | 0.143±0.079 | 0.054±0.028 | 0.215±0.032 | 0.382±0.034 |

Table 24: Results for different values of Z on Stock 72 dataset.

| | $Z\_2\_3\_3$ | $Z\_4\_6\_6$ | $Z\_10\_15\_15$ | $Z\_40\_60\_60$ | $Z\_100\_150\_150$ |
|---|---|---|---|---|---|
| Discr-score | 0.046±0.032 | 0.044±0.017 | 0.028±0.027 | 0.025±0.019 | 0.035±0.034 |
| Pred-score | 0.186±0.002 | 0.184±0.001 | 0.185±0.001 | 0.185±0.001 | 0.185±0.001 |

## K.3 Train and Test Split on Stock Data

In Table 25 we show the evaluation of the approaches while randomly splitting data between training set (i.e., 80% of dataset) and test set (i.e., 20% of dataset). The table reports the quantitative metrics for the regular Stock data with 24 length, while Figure 17 shows the t-SNE visualization (Top) and the data distribution (Bottom). These results confirm the results from the our main paper, with iHT achieving the best performance in discriminative and predictive scores and second best performance in Marginal Score. Additionally, given that the diversity of generated time series is an important aspect of the generation task, we use a novel metric to quantitatively evaluate the diversity of time series, namely *sym*-Recall (Alaa et al., 2021) which extends the $\beta$-Recall metric from (Khayatkhoei & AbdAlmageed, 2023). This metric quantifies the diversity of time series as the extent to which synthetic samples cover the full variability of real samples. The results are shown in the last row of Table 25. We observe that iHT shows the second best performance with regards to diversity, with DiffTime showing the best performance in diversity.

Table 25: Generation performance on regular Stock24 dataset

| Metric | iHT (Ours) | GT-GAN | TimeGAN | DiffTime | LS4 | FFlows |
|---|---|---|---|---|---|---|
| Discr-score ↓ | 0.054±0.028 | 0.273±0.046 | 0.068±0.018 | 0.079±0.014 | 0.154±0.080 | 0.426±0.032 |
| Pred-score ↓ | 0.037±0.000 | 0.046±0.001 | 0.043±0.001 | 0.044±0.001 | 0.039±0.000 | 0.055±0.003 |
| Marginal Score ↓ | 0.355 | 0.403 | 0.434 | 0.359 | 0.513 | 0.335 |
| *sym*-Recall ↑ | 0.594 | 0.500 | 0.413 | 0.787 | 0.206 | 0.000 |

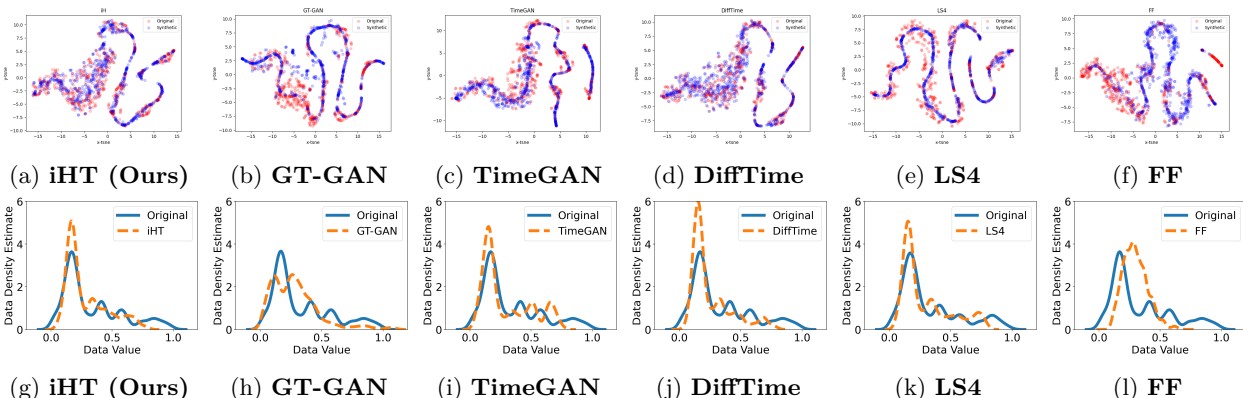

(a) **iHT (Ours)**  (b) **GT-GAN**  (c) **TimeGAN**  (d) **DiffTime**  (e) **LS4**  (f) **FF**

(g) **iHT (Ours)**  (h) **GT-GAN**  (i) **TimeGAN**  (j) **DiffTime**  (k) **LS4**  (l) **FF**

Figure 17: (Top) t-SNE visualizations on regular Stock24 data where a greater overlap of blue and red dots shows a better distributional-similarity between the generated data and original data. (Bottom) Data distribution on regular Stock24 data. Our approach shows the best performance.

## L  Visualizations with tSNE and data distributions

In this section we report an additional evaluation of the synthetic and real data distributions. We evaluate the synthetic distributions on Stock24, including missing data from 30% to 70%; then we analyse synthetic data on longer stock time-series, i.e., stock72 and stock360; and finally we evaluate the synthetic distributions for Energy data, which has 28 dimensions.

### L.1  Data distribution

First we plot the real (blue) and synthetic (orange) distributions empirically evaluated through a kernel-density estimation of real and generated data. Figure 18 shows the empirical distributions for Stock24, where iHT has among the closest match with original data. The superior performance of iHT is more evident with irregular data, from Figure 19 to Figure 21, where iHT is always able to closely resemble the real data distributions.

The performance of iHT is consistent with longer time-series (Figure 22 and Figure 23) and highly dimensional data like Energy in Figure 24.

### L.2  tSNE visualization

We now evaluate the real (blue) and synthetic (red) distributions through t-SNE visualizations. Figure 25 shows the t-SNE plots for Stock24, where iHT has among the best performance (i.e., the synthetic data almost completely overlap with real data). As mentioned in the previous section, the superior performance of iHT is more evident with irregular data, from Figure 26 to Figure 28, where iHT is the only method to closely resemble the real data distribution. While GT-GAN reproduces similarly the real data distributions, the synthetic distributions are more condensed around the original data, and don't cover the full space.

For longer time-series (Figure 29 and Figure 30) the t-SNE plots show that iHT is able to better reproduce the original data distributions. Finally, Figure 31 shows the t-SNE plots for energy data where with consistent performance for iHT.

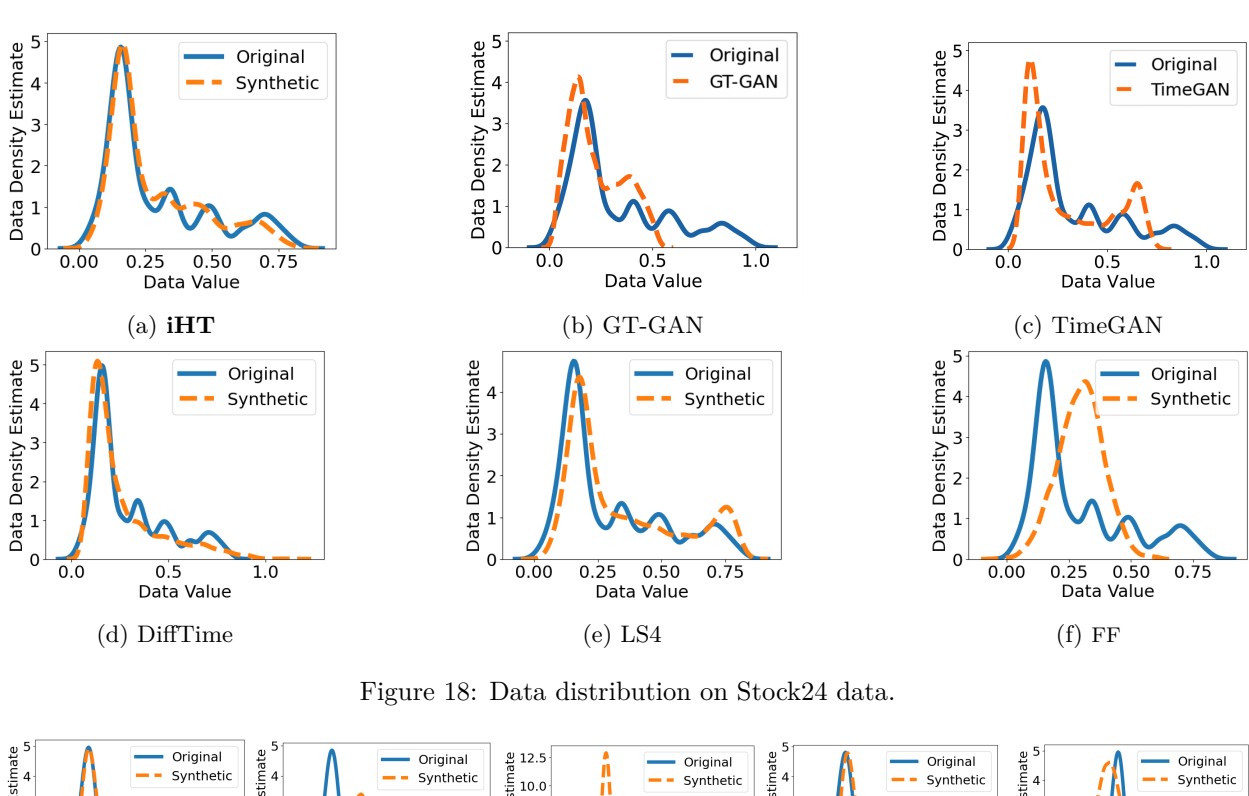

Figure 18: Data distribution on Stock24 data.

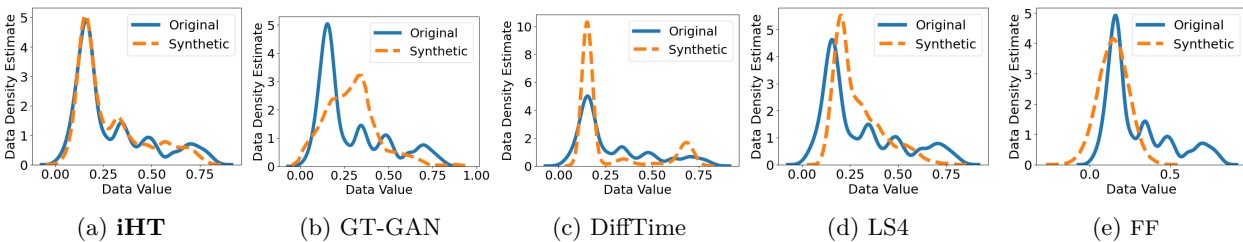

Figure 19: Data distribution on irregular Stock24 data (Missing 70%).

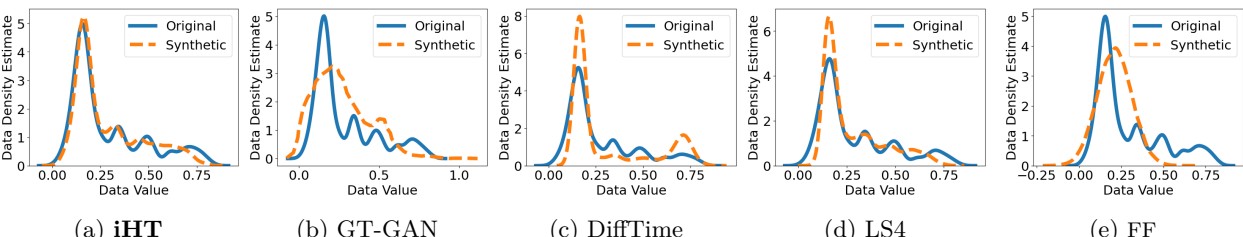

Figure 20: Data distribution on irregular Stock24 data (Missing 50%).

Figure 21: Data distribution on irregular Stock24 data (Missing 30%).

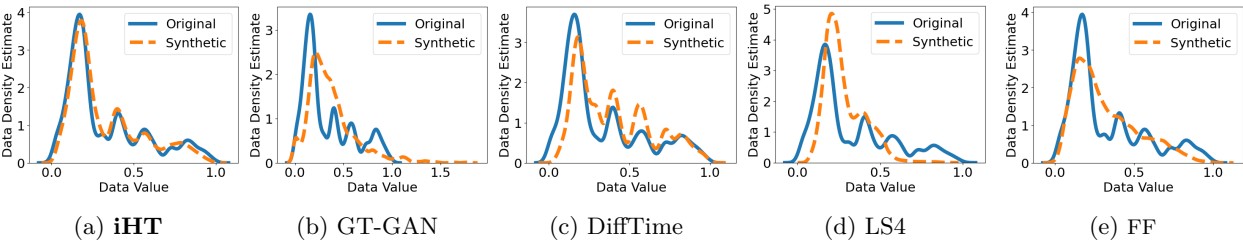

Figure 22: Data distribution on regular Stock72 data.

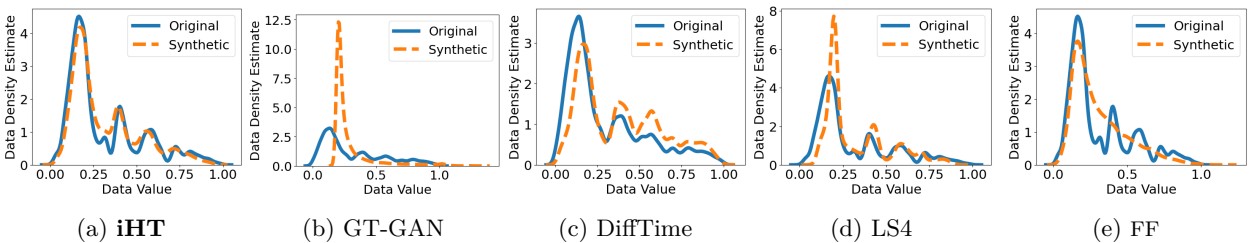

Figure 23: Data distribution on regular Stock360 data.

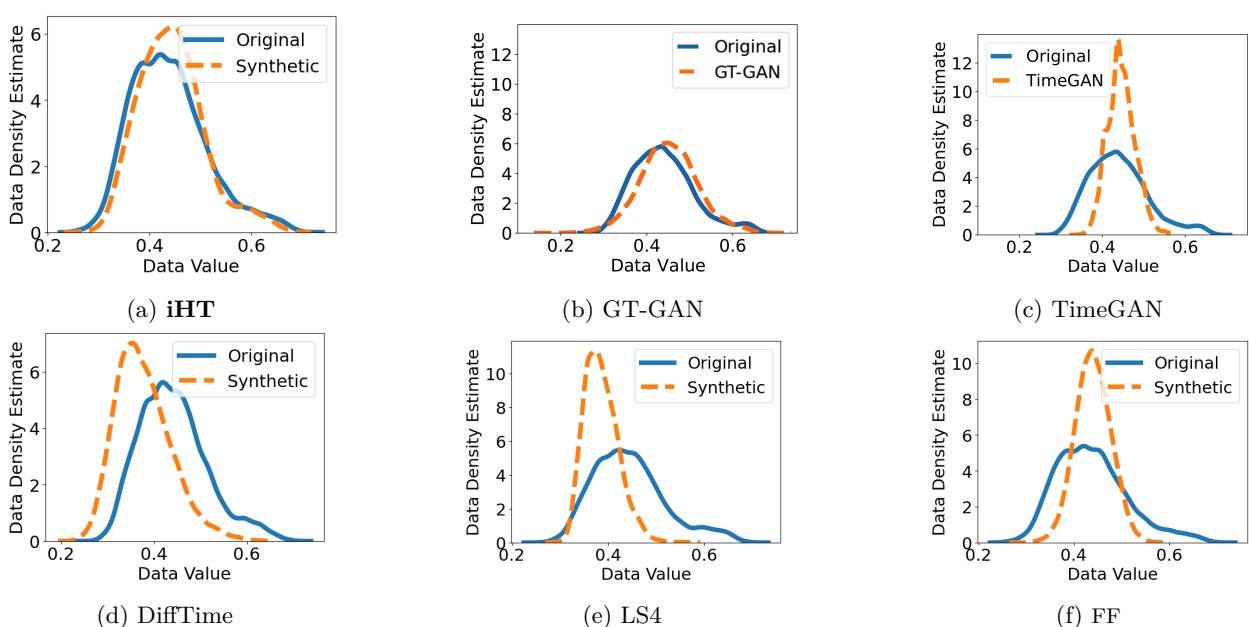

Figure 24: Data distribution on Energy data.

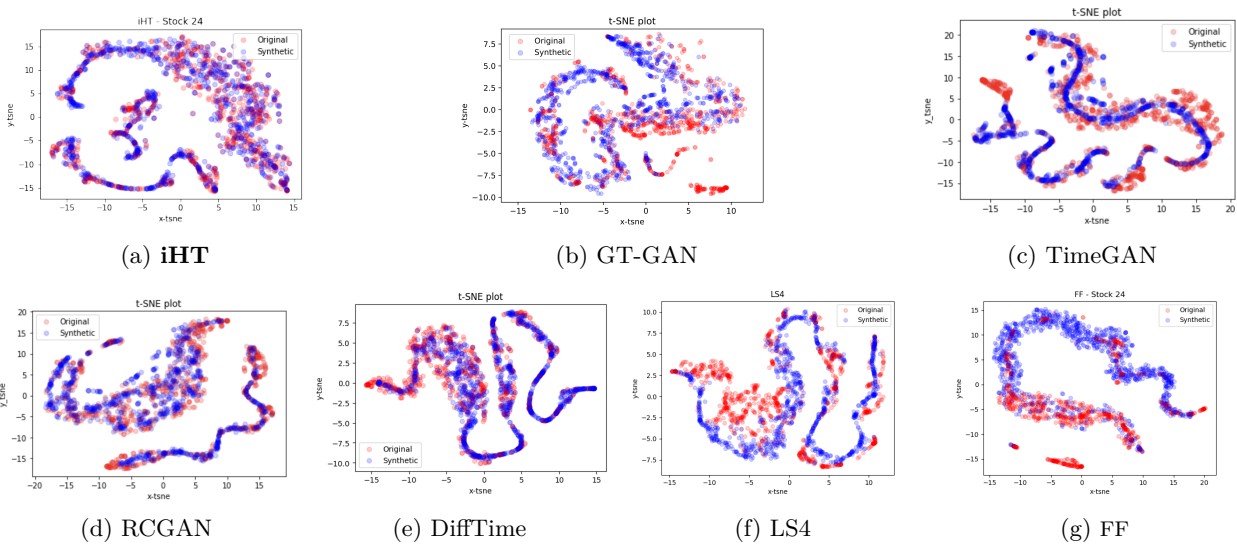

(a) **iHT**       (b) GT-GAN       (c) TimeGAN

(d) RCGAN     (e) DiffTime     (f) LS4     (g) FF

Figure 25: t-SNE visualizations on Stock24 data, where a greater overlap of blue and red dots shows a better distributional-similarity between the generated data and original data. Our approach shows the best performance.

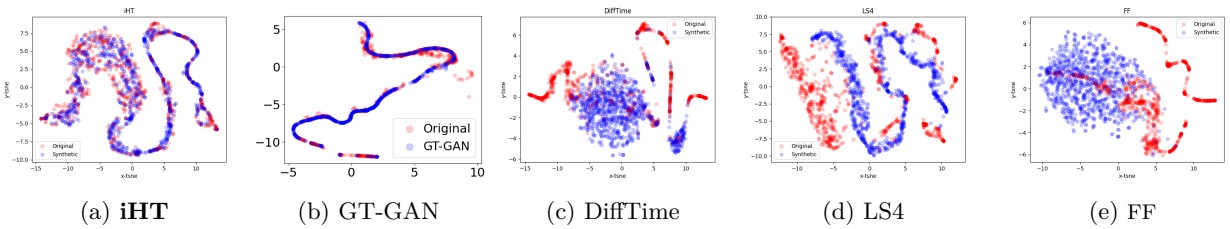

(a) **iHT**     (b) GT-GAN     (c) DiffTime     (d) LS4     (e) FF

Figure 26: t-SNE visualizations on irregular Stock24 data (Missing 70%), where a greater overlap of blue and red dots shows a better distributional-similarity between the generated data and original data. Our approach shows the best performance.

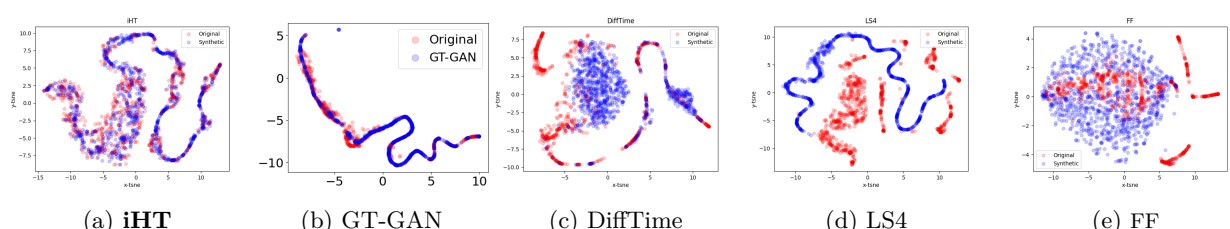

(a) **iHT**     (b) GT-GAN     (c) DiffTime     (d) LS4     (e) FF

Figure 27: t-SNE visualizations on irregular Stock24 data (Missing 50%), where a greater overlap of blue and red dots shows a better distributional-similarity between the generated data and original data. Our approach shows the best performance.

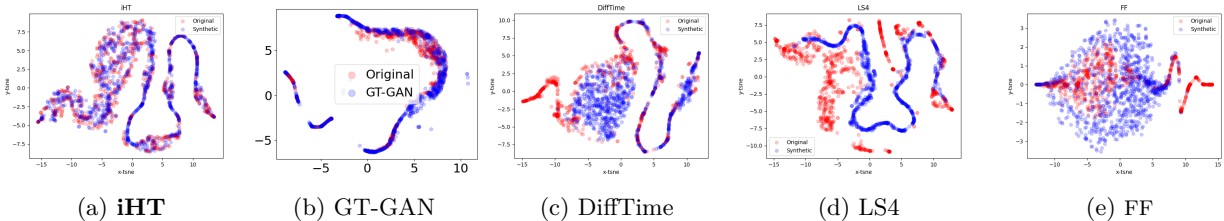

(a) **iHT**     (b) GT-GAN     (c) DiffTime     (d) LS4     (e) FF

Figure 28: t-SNE visualizations on irregular Stock24 data (Missing 30%), where a greater overlap of blue and red dots shows a better distributional-similarity between the generated data and original data. Our approach shows the best performance.

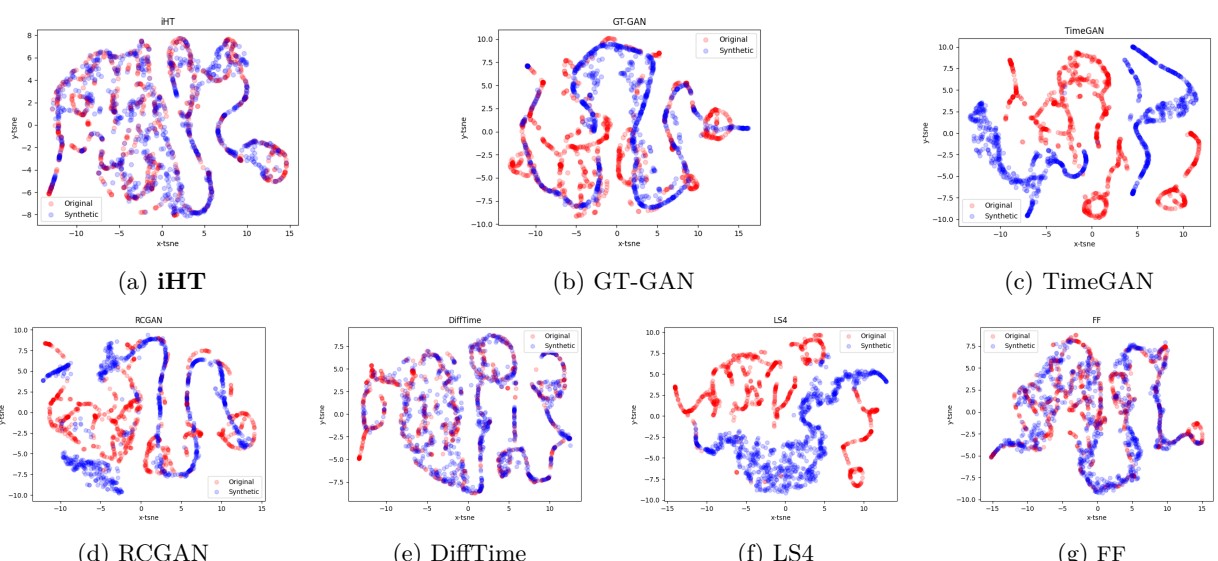

(a) **iHT**     (b) GT-GAN     (c) TimeGAN

(d) RCGAN     (e) DiffTime     (f) LS4     (g) FF

Figure 29: t-SNE visualizations on Stock72 data, where a greater overlap of blue and red dots shows a better distributional-similarity between the generated data and original data. Our approach shows the best performance.

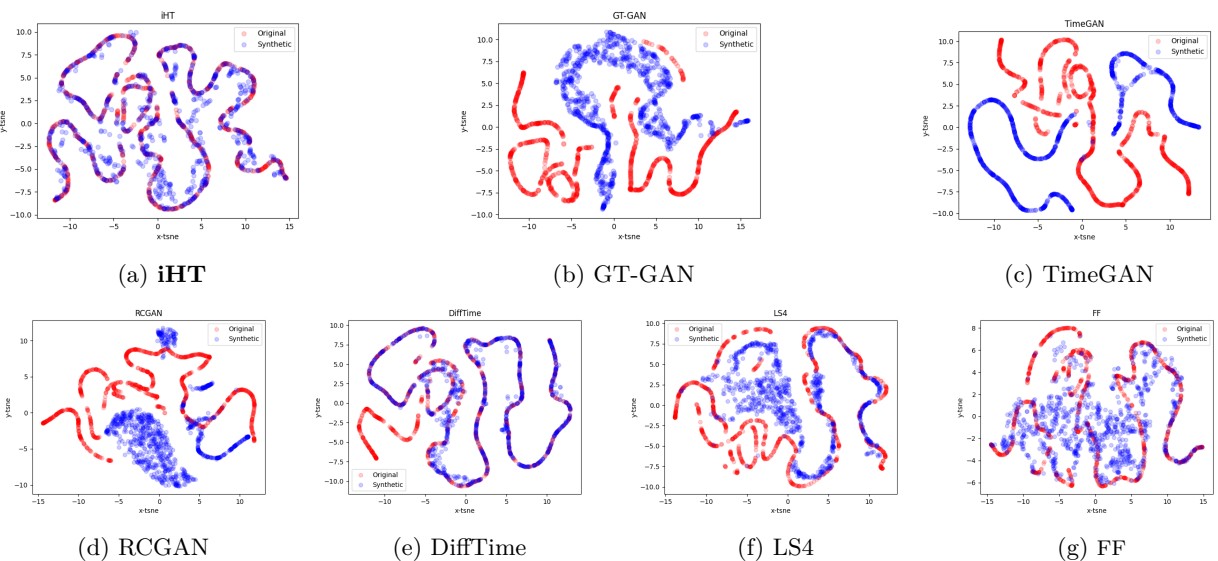

Figure 30: t-SNE visualizations on Stock360 data, where a greater overlap of blue and red dots shows a better distributional-similarity between the generated data and original data. Our approach shows the best performance.

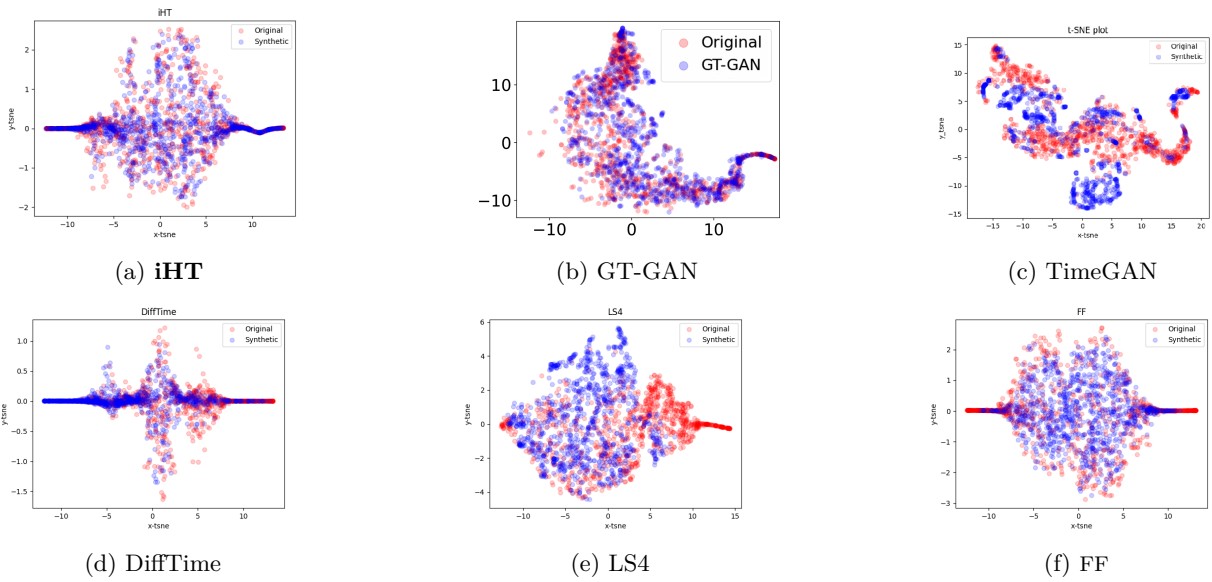

Figure 31: t-SNE visualizations on Energy data, where a greater overlap of blue and red dots shows a better distributional-similarity between the generated data and original data. Our approach shows the best performance.

# M  Financial Returns and Autocorrelation

We now evaluate the distributions of two well known properties (i.e., *stylized facts*) of financial time-series, namely the returns and autocorrelation. We consider stock uni-variate data with length of 72. Figure 32 and Figure 33 show the *returns* and the *autocorrelation of returns* distributions, respectively. The two figures confirm the ability of iHT to learn the real data properties (i.e., the real and synthetic distributions mostly overlap).

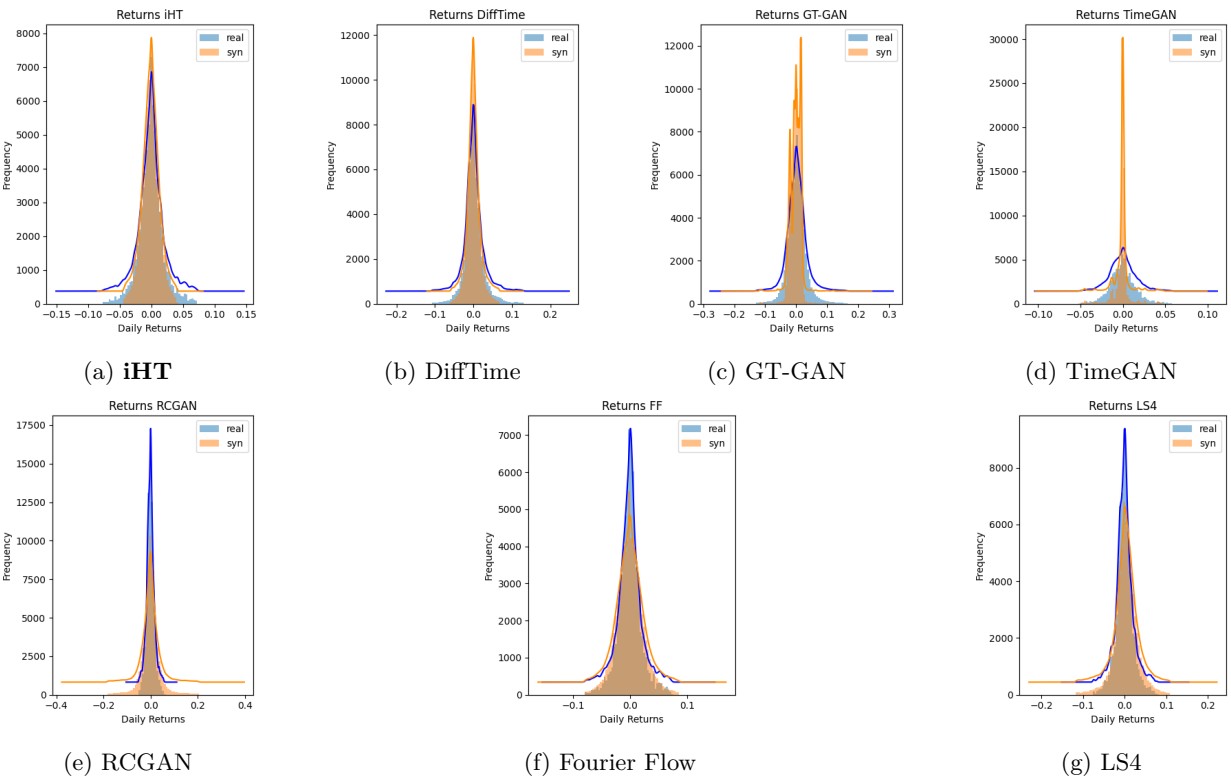

(a) **iHT**    (b) DiffTime    (c) GT-GAN    (d) TimeGAN

(e) RCGAN    (f) Fourier Flow    (g) LS4

Figure 32: Returns distribution of stock time-series with length 72.

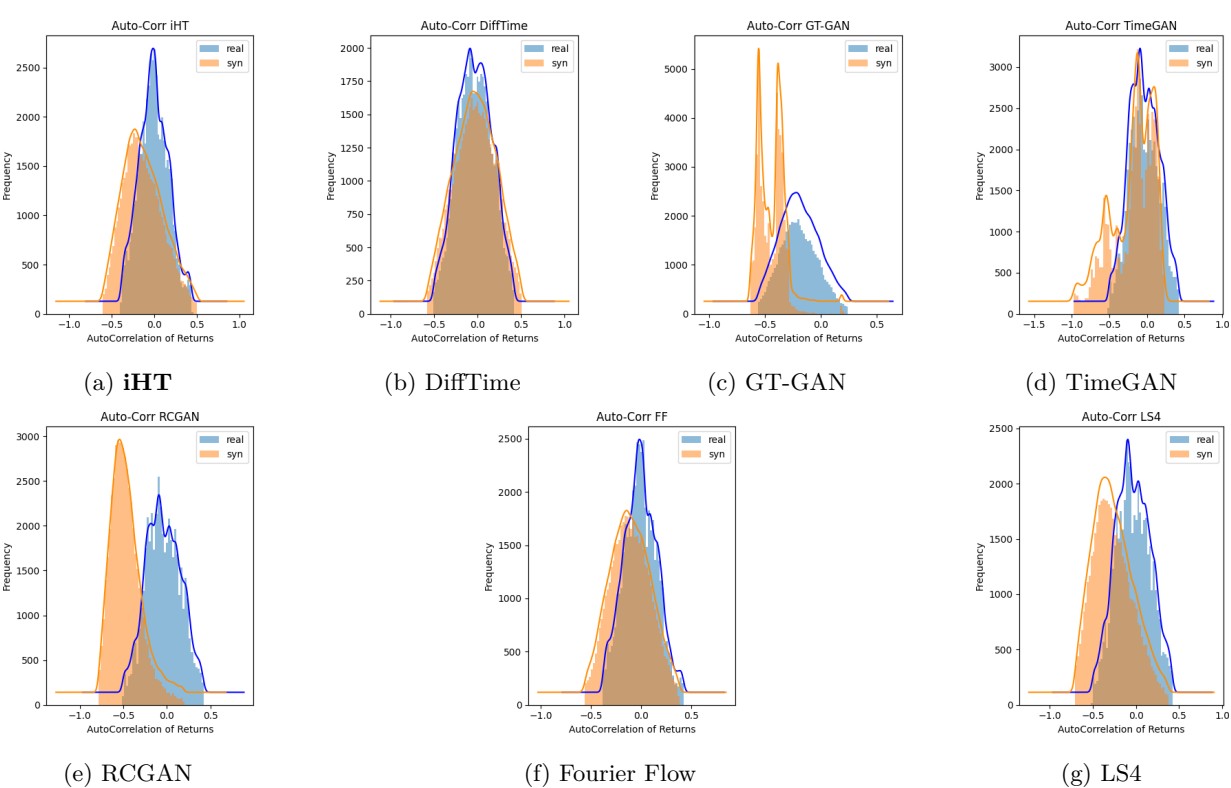

Figure 33: Autocorrelation of returns distributions of stock time-series with length 72.

# N  Visualizations of Generated Samples

Finally we report examples of generated time-series. It is worth to mention that in Figure 34 the synthetic time-series from iHT effectively respect the open-high-low-close relationship from the real data – high (low) is the highest (lowest) series. Such data property is preserved also when the model is trained on missing data, as shown in Figure 35, Figure 36, and Figure 37. With the exception of DiffTime, which is specifically designed for constrained time-series generation, most of the existing approaches do not preserve such property.

Energy data is shown in Figure 38 for regular time-series, and in Figure 39, Figure 40, and Figure 41 for irregular time-series. Considering that Energy has 28 features, with different scales, we plot only the first 5 normalized features.

Finally, we plot longer-times for regular stock72 in Figure 42. While we plot the irregular stock72 in Figure 43, Figure 44, and Figure 45.

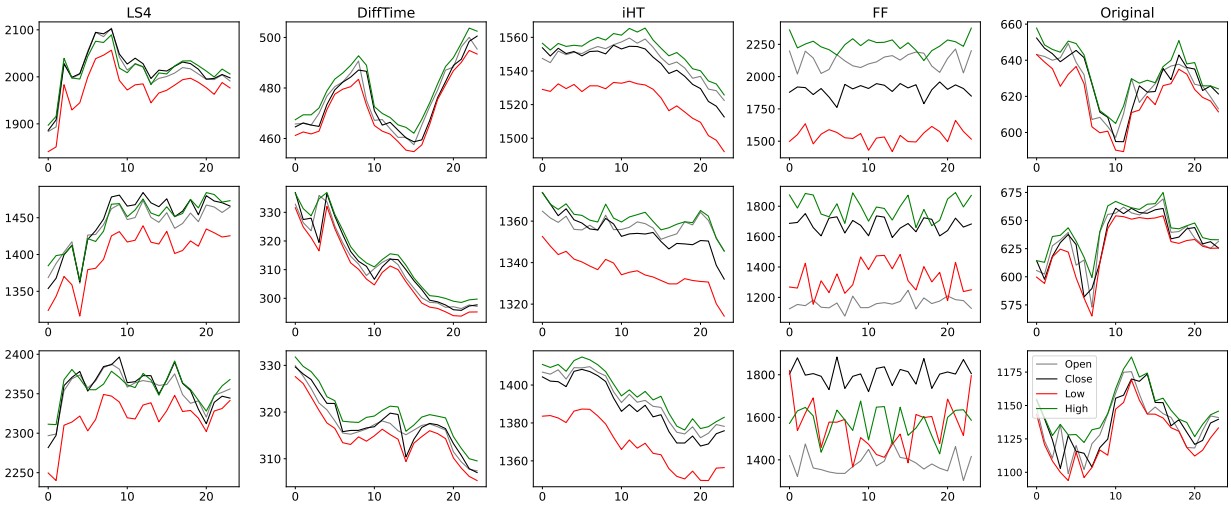

Figure 34: An example of Regular Stock24 samples.

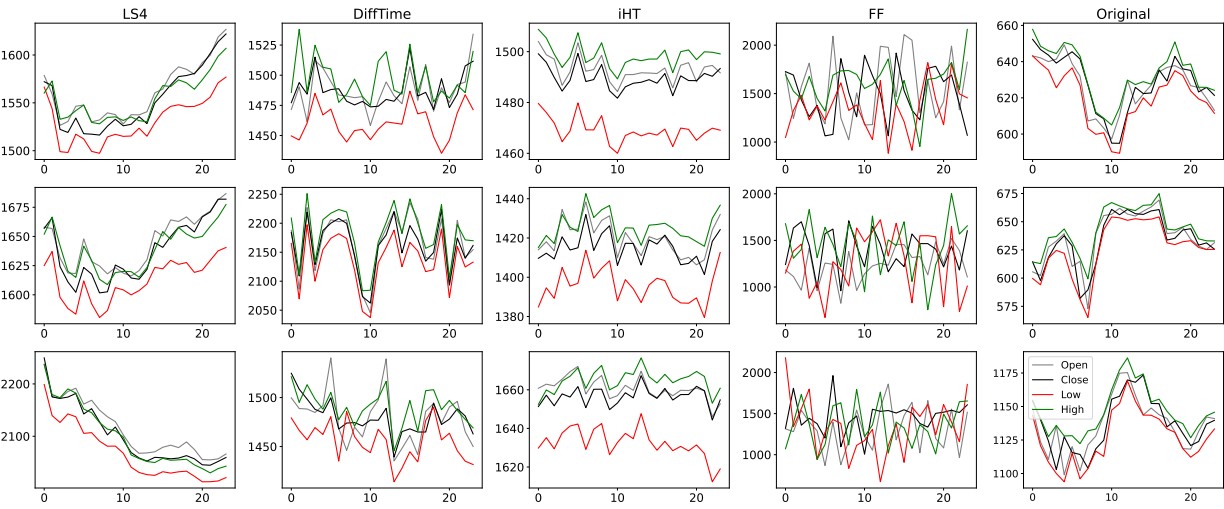

Figure 35: An example of Irregular Stock24 samples (70% missing data).

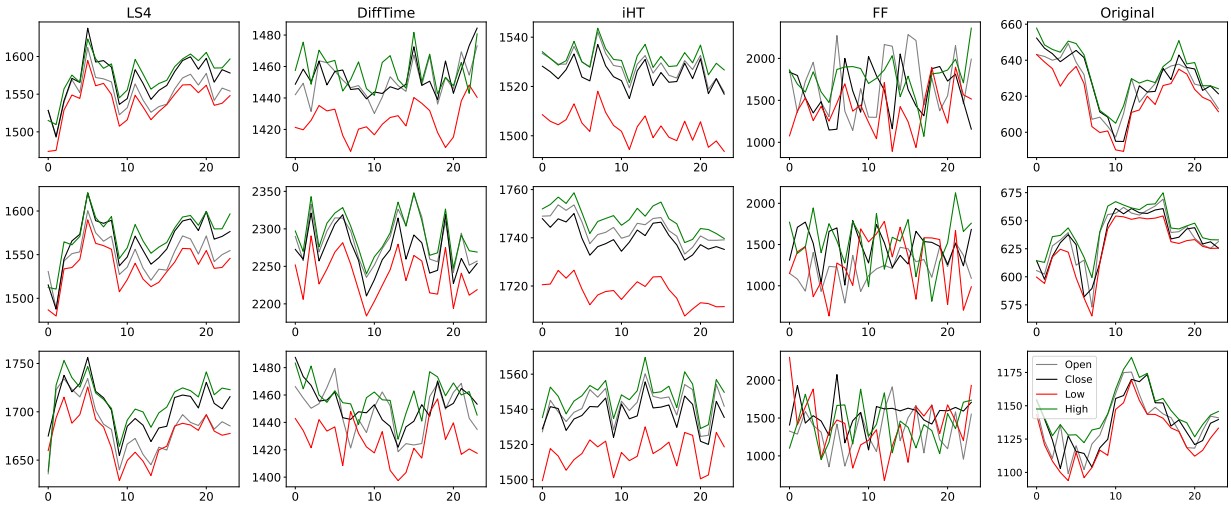

Figure 36: An example of Irregular Stock24 samples (50% missing data).

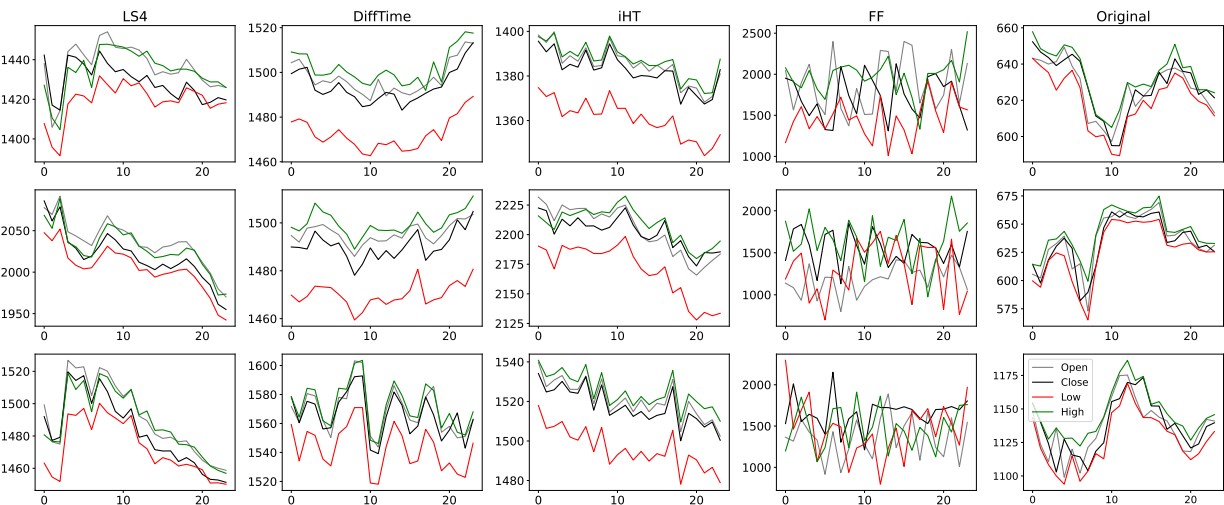

Figure 37: An example of Irregular Stock24 samples (30% missing data).

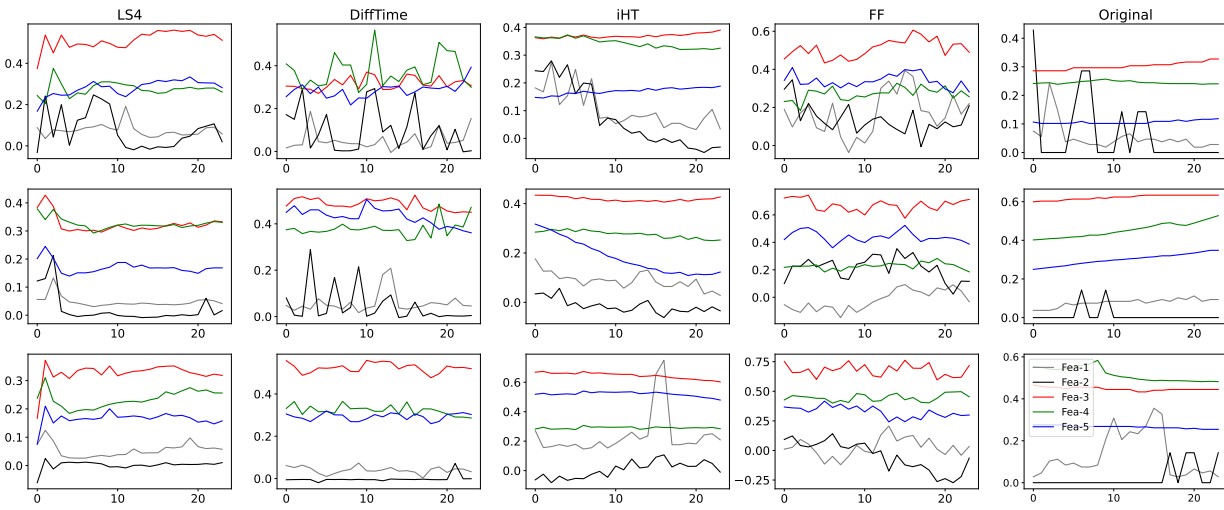

Figure 38: An example of Regular Energy24 samples.

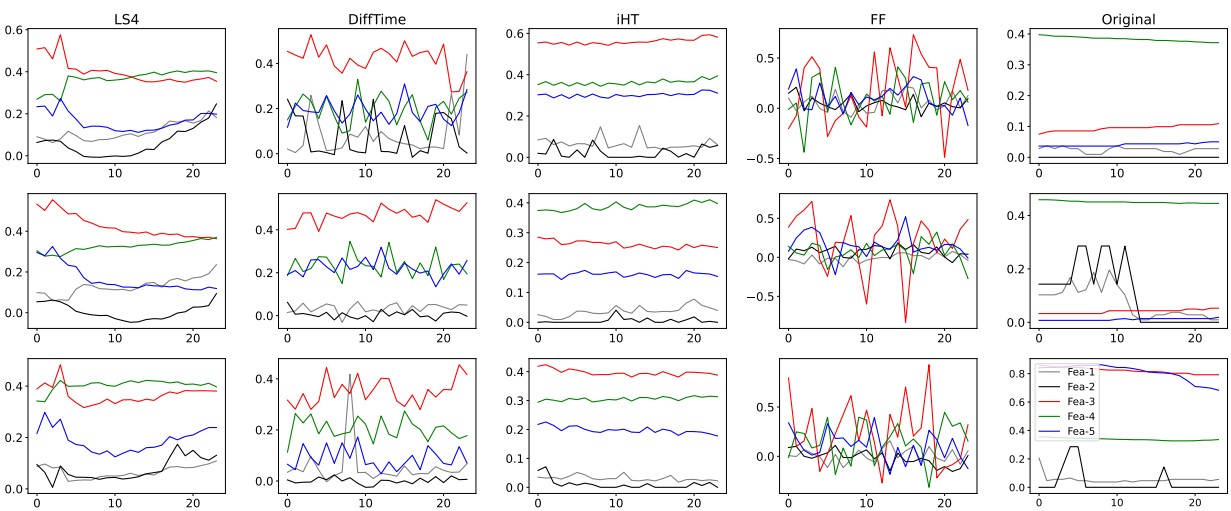

Figure 39: An example of Irregular Energy24 samples (70% missing data).

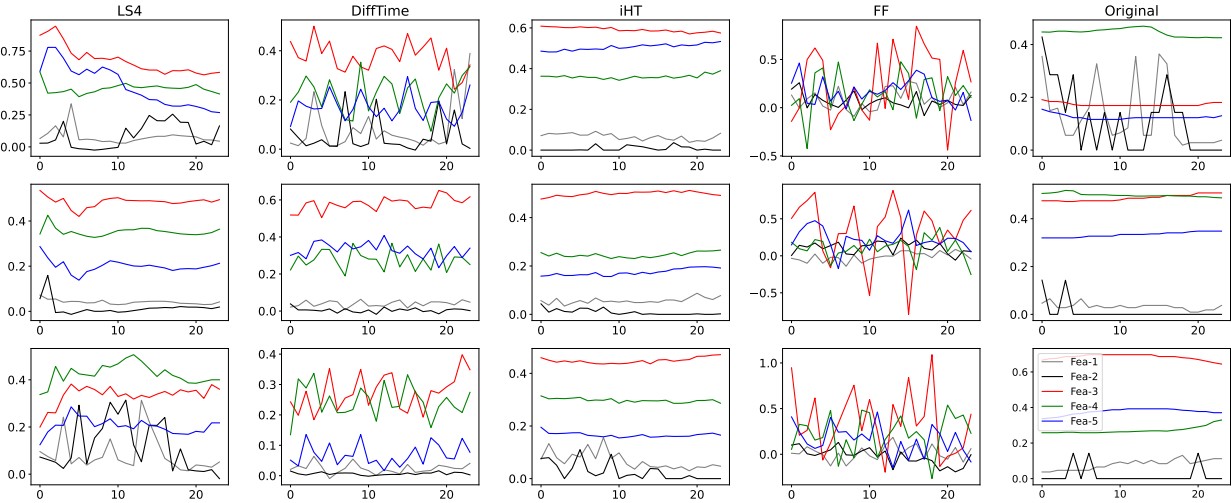

Figure 40: An example of Irregular Energy24 samples (50% missing data).

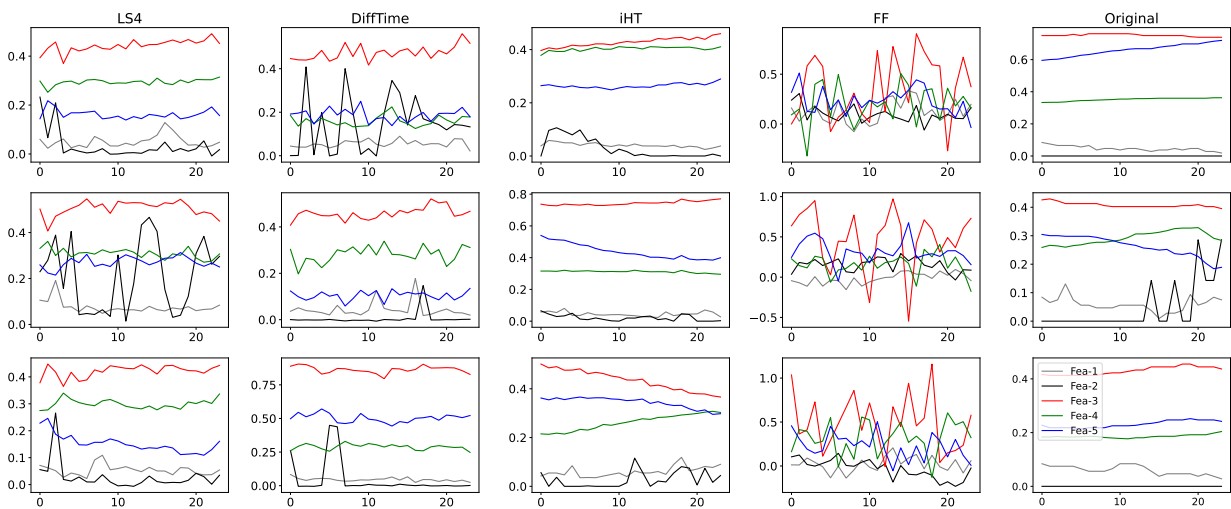

Figure 41: An example of Irregular Energy24 samples (30% missing data).

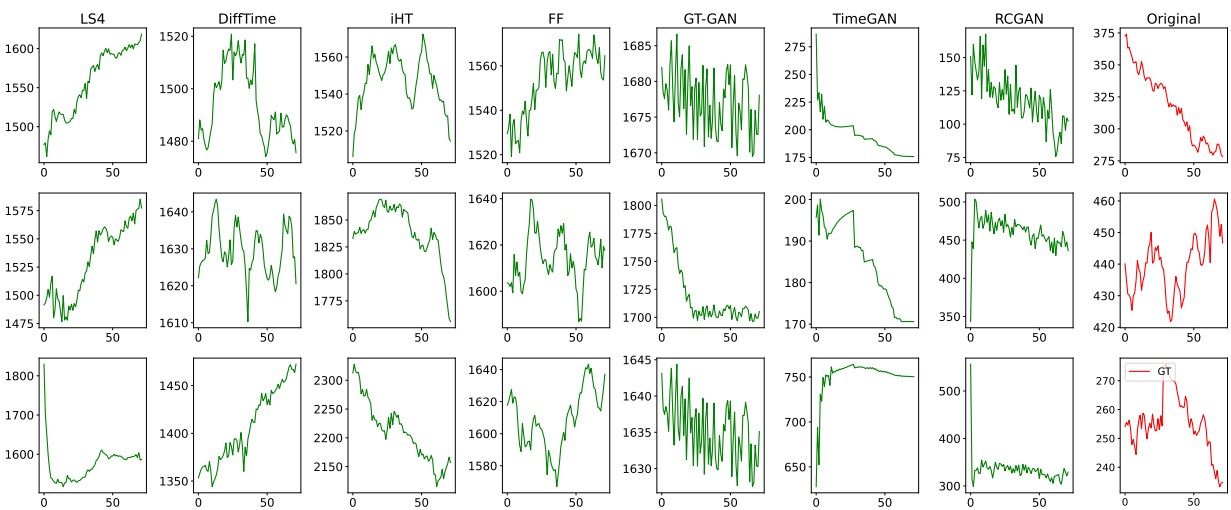

Figure 42: An example of Regular Stock72 samples.

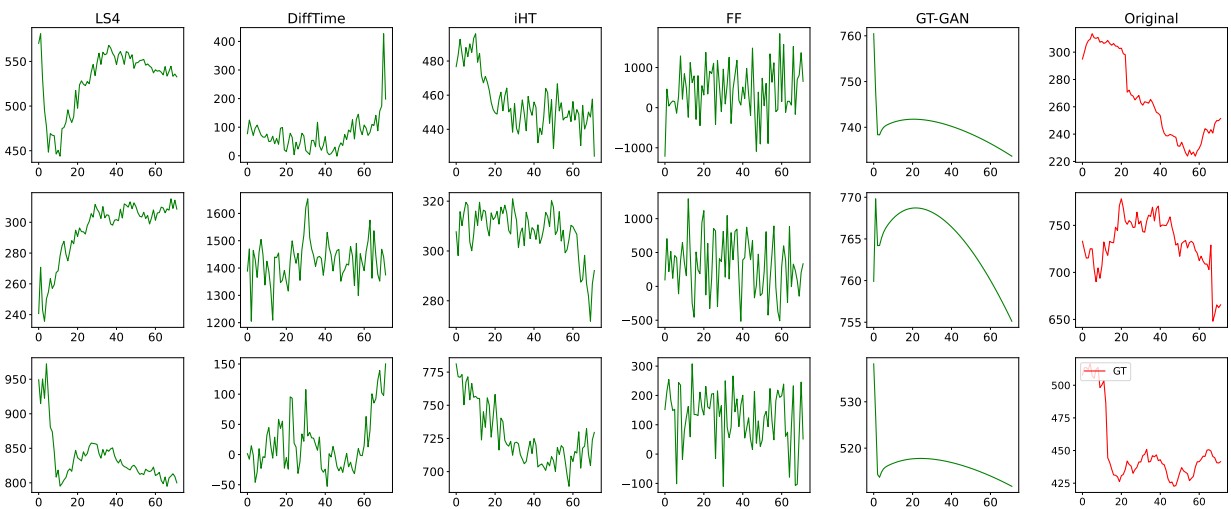

Figure 43: An example of Irregular Stock72 samples (70% missing data).

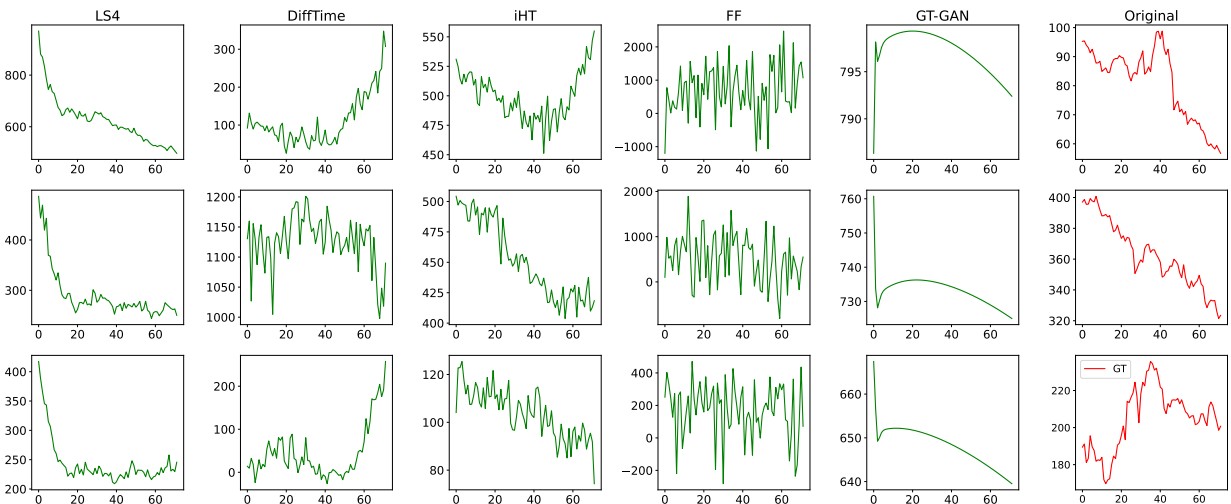

Figure 44: An example of Irregular Stock72 samples (50% missing data).

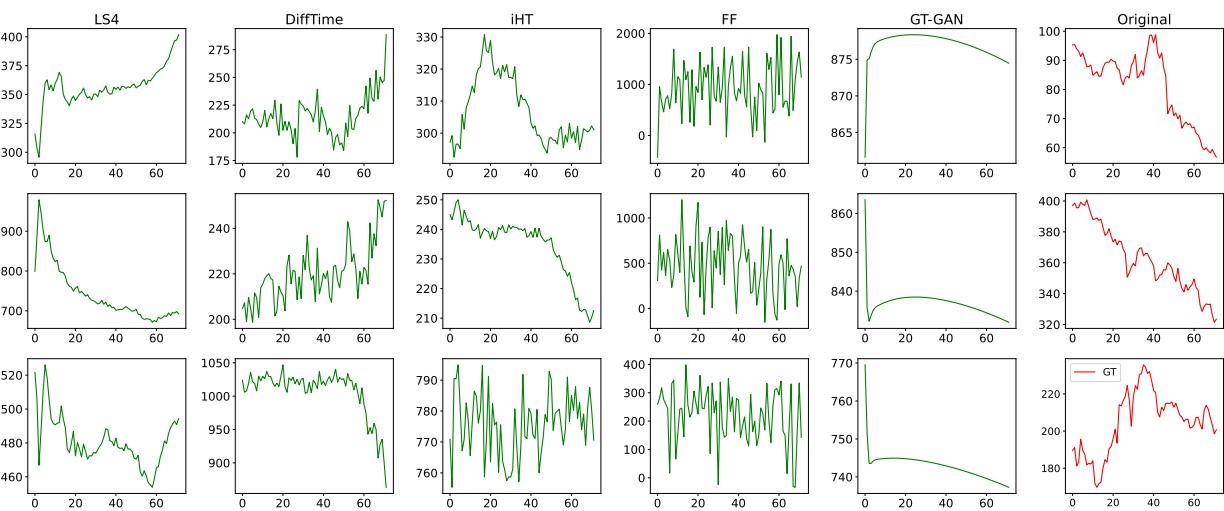

Figure 45: An example of Irregular Stock72 samples (30% missing data).

## Supplemental References

Ahmed M. Alaa, Boris van Breugel, Evgeny S. Saveliev, and Mihaela van der Schaar. How faithful is your synthetic data? sample-level metrics for evaluating and auditing generative models. In *Proceedings of the International Conference on Machine Learning (ICML)*, 2021.

Nuri Benbarka, Timon Höfer, Hamd ul Moqeet Riaz, and Andreas Zell. Seeing implicit neural representations as fourier series. pp. 2283–2292, 2022.

Andrew Brock, Theo Lim, J.M. Ritchie, and Nick Weston. SMASH: One-shot model architecture search through hypernetworks. In *Proceedings of the International Conference on Learning Representations (ICLR)*, 2018.

Robin John Hyndman and George Athanasopoulos. *Forecasting: Principles and Practice*. OTexts, Australia, 2nd edition, 2018.

Mahyar Khayatkhoei and Wael AbdAlmageed. Emergent asymmetry of precision and recall for measuring fidelity and diversity of generative models in high dimensions. In *Proceedings of the International Conference on Machine Learning (ICML)*, 2023.

Jeong Joon Park, Peter Florence, Julian Straub, Richard Newcombe, and Steven Lovegrove. Deepsdf: Learning continuous signed distance functions for shape representation. In *Proceedings of the IEEE Conference on Computer Vision and Pattern Recognition (CVPR)*, June 2019.

Jacob Russin Russin, Randall O'Reilly, and Yoshua Bengio Bengio. Deep learning needs a prefrontal cortex. In *Bridging AI and Cognitive Science ICLR 2020 Workshop*, 2020.

Andrei A. Rusu, Dushyant Rao, Jakub Sygnowski, Oriol Vinyals, Razvan Pascanu, Simon Osindero, and Raia Hadsell. Meta-learning with latent embedding optimization. In *Proceedings of the International Conference on Learning Representations (ICLR)*, 2019.

Kenneth O. Stanley, David B. D'Ambrosio, and Jason Gauci. A Hypercube-Based Encoding for Evolving Large-Scale Neural Networks. *Artificial Life*, 15(2):185–212, 04 2009.

Matthew Tancik, Ben Mildenhall, Terrance Wang, Divi Schmidt, Pratul P. Srinivasan, Jonathan T. Barron, and Ren Ng. Learned initializations for optimizing coordinate-based neural representations. In *Proceedings of the IEEE Conference on Computer Vision and Pattern Recognition (CVPR)*, 2021.

Yusuke Tashiro, Jiaming Song, Yang Song, and Stefano Ermon. Csdi: Conditional score-based diffusion models for probabilistic time series imputation. *Advances in Neural Information Processing Systems (NeurIPS)*, 2021.

Johannes von Oswald, Christian Henning, Benjamin F. Grewe, and João Sacramento. Continual learning with hypernetworks. In *Proceedings of the International Conference on Learning Representations (ICLR)*, 2020.

Xiaozhe Wang, Kate Smith, and Rob Hyndman. Characteristic-based clustering for time series data. *Data Min. Knowl. Discov.*, 13(3):335–364, nov 2006.

Chris Zhang, Mengye Ren, and Raquel Urtasun. Graph hypernetworks for neural architecture search. In *Proceedings of the International Conference on Learning Representations (ICLR)*, 2019.

Dominic Zhao, Seijin Kobayashi, João Sacramento, and Johannes von Oswald. Meta-learning via hypernetworks. In *4th Workshop on Meta-Learning at NeurIPS 2020 (MetaLearn 2020)*. Advances in Neural Information Processing Systems (NeurIPS), 2020.

