# OpenReview forum: "iHyperTime: Interpretable Time Series Generation with Implicit Neural Representations"
_TMLR — Accepted by TMLR_

### Review · Reviewer_Gp9n · 2024-03-15

**Summary Of Contributions:**

Based on TSNet, and INR, this paper proposes a new method for time series generation.  The experiments are conducted on multiple dataset with varying characteristics. The results show that the proposed method achieves better performance than the existing methods.

**Audience:**

Yes

**Broader Impact Concerns:**

No.

**Claims And Evidence:**

Yes

**Requested Changes:**

See the weaknesses.

**Strengths And Weaknesses:**

Strengths:

-This paper proposes a new architecture, that combine the advantage of both TSNet and INR for the  trend-seasonality time
series representation.  the proposed method has stronger interpretability, as it points the information flow process in different networks.

-The authors conduct extensive experiments on various benchmarks.  On all these datasets, it shows superiority on multiple benchmarks. The method also has strong generalization ability considering the diversity of these datasets.

Weaknesses:

-The proposed method combine the existing architectures to improve performance. It is better to further discuss the difference compared with TSNet and INR, which can further enhance the quality of this paper.

-The authors propose three modules, trend hypernet, seasonal hypernet, and residual hypernet, which have different functions. How to validate that the practical function in a trained network? More evidences are required.

---

> ### Author Response · Authors · 2024-06-12
>
> We thank the reviewer for their valuable feedback. We hope they find our answers useful, and we are happy to answer any additional questions.
>
> **W1.**
> We acknowledge the importance of clarifying the distinctions between our proposed iHyperTime architecture and the underlying TSNet and INR components. We have revised the manuscript to provide a more in-depth discussion of these differences. Specifically, we have expanded the discussion in Section 3.1. In this discussion we emphasize that TSNet is not merely an application of existing INR techniques to time series data. It is a novel INR architecture specifically designed for time series representation and decomposition. The incorporation of trend, seasonality, and residual blocks, along with the use of polynomial and Fourier basis functions, enables TSNet to provide an interpretable and disentangled representation of time series data. This interpretability is a key advantage of TSNet over generic INRs, which typically lack such explicit decomposition capabilities. For clarity, we have highlighted the expanded discussion in blue in the new manuscript.
>
> **W2.**
> We would like to thank the reviewer for the insightful question regarding the validation of the practical functions of the trend, seasonal, and residual hypernets in our iHyperTime architecture. We have conducted several analyses within the scope of this work that provide evidence for their distinct roles:
> * Experiments on synthetic data (Section 4.6 and Appendix G): Our experiments on synthetic datasets with known ground-truth components demonstrate that each hypernet effectively learns to represent its intended pattern. The trend hypernet captures slow-varying behavior, the seasonal hypernet captures periodic fluctuations, and the residual hypernet captures the remaining high-frequency components. These experiments also include a comparison with the well-established STL decomposition method, and t-SNE visualizations that reveal distinct clusters corresponding to different trend directions and seasonal frequencies.
> * Analysis of real-world data (Appendix H): We compare the Trend-Seasonality-Residuals components produced by iHyperTime with those obtained from the STL decomposition method on real-world datasets with known strong trend and seasonality components. The observed agreement between the two methods further supports the effectiveness of our hypernets in capturing real-world trend and seasonality patterns.
>  * Controlled generation experiments (Appendix I): We have conducted experiments where we manipulate the input to the _Trend Hypernet_ to generate time series with specific trend patterns. The results demonstrate that the generated time series closely follow the desired trends, providing additional evidence of the trend hypernet's ability to control the trend component of the generated data.
>
> We believe these existing analyses provide compelling evidence that the trend, seasonal, and residual hypernets in iHyperTime effectively fulfil their intended functions, capturing and controlling the corresponding components of time series data in both synthetic and real-world scenarios.

---

### Review · Reviewer_2zXs · 2024-04-18

**Summary Of Contributions:**

The authors propose a novel method of time series data generation using a hypernetwork iHyperTime that is responsible for predicting the weights of Implicit Neural Representation of the time series data, namely TSNet. Through extensive experimentation the authors show that the proposed method performs better than the existing method specifically for the time series which contain missing data.

**Audience:**

Yes

**Claims And Evidence:**

Yes

**Requested Changes:**

In general, I found that the Experiment section discusses the setup used for experimentation in details along with the tables for the various experiments that have been carried out. But the discussion lacks the insights that one can draw from these tables. In other words the interpretation of the tables is left to the reader (As pointed out in W2 and W3 above). I would request authors to make these inferences explicit in the text discussing the tables wherever possible.

Please fix or explain the weaknesses I mentioned above.

**Strengths And Weaknesses:**

**Strengths**

1. Using INR’s resolution agnostic property for handling irregularly sampled or long series data.

2. Presentation and readability are good.

3. The proposed method is capable of handling irregular as well as long time series.

**Weakness:**

W1. The authors mention that “The use of a set encoder introduces permutation invariance in the computation”. Permutation invariance: Is it desirable for time series data where serial dependencies are present?

W2. In Table 1, why does the predictive score drops by a magnitude of 10 (tables 2, 3, 4) in case of missing data for “Energy24” dataset whereas for all other datasets the values remain comparable to the values in Table 1?

W3. The insights and trends from Table 7 are not clear to me. It appears that all variants are equally useful (indicated by the predictive scores) however, at the same time the discriminative scores indicate that the iHT (no FFT) and iHT-SIREN variants are not realistic enough? This seems to be contradictory. Please clarify.

W4. I think the authors should perform the ablation of Table 7 with missing data as well. I believe there the difference between will be more significant allowing us to analyze the role of these contribution in case of missing data.

---

> ### Author Response · Authors · 2024-06-12
>
> We thank the reviewer for the insightful comments and suggestions. We hope they find our answers useful, and we are happy to answer any additional questions.
>
> **W1.**
> Thank you for your insightful comment. While our set encoder is permutation invariant, the temporal information is conserved because the time coordinate is explicitly included in each set element alongside its corresponding value. This allows subsequent layers to learn temporal relationships, aligning with approaches like "Set Functions for Time Series" [1].
> Additionally, our methodology is consistent with the observations made in "Order Matters" [2], suggesting that encoding sequences as sets can offer advantages in certain contexts.
>
> [1]  Horn, M., Moor, M., Bock, C., Rieck, B. and Borgwardt, K., (2020). Set Functions for Time Series. ICML 2020.
> [2]  Vinyals, O., Bengio, S. and Kudlur, M., (2016). Order Matters: Sequence to sequence for sets. ICLR 2016
>
> **W2.**
> Thank you for your careful attention to the results presented in our tables and for pointing out the discrepancy in the predictive scores for the "Energy24" dataset between Table 1 and Tables 2, 3, and 4. This difference arises due to using different evaluation schemes commonly employed in the literature, which differ for regular and irregular time series.
>
> In Table 1, we follow the predictive score definition from TimeGAN, which predicts only one future element based on past elements. This aligns with their experimental setup and ensures a fair comparison in the regular time series scenario.
>
> However, for the irregular time series experiments presented in Tables 2, 3, and 4, we adopt the predictive task proposed by GT-GAN.
> This predictive task involves forecasting the entire future vector, rather than just a single element from a given channel. Irregular time series are often sampled at random time steps and channels, and predicting only one future element can make the evaluation less reliable. We opted for the GT-GAN evaluation methodology for irregular time series as it provides a more comprehensive assessment of the model's ability to capture complex temporal dependencies in the presence of irregularly sampled data and it ensures a fairer comparison between models specifically designed for irregular time series generation.
>
> In the case of Energy24, which contains 28 channels, the more challenging task of predicting all channels instead of just one may result in a lower predictive score (better performance) compared to the TimeGAN evaluation. This is because the error is averaged over all 28 channels, potentially masking poor performance on individual channels. This effect is also observed, although to a lesser extent, in the Stock24 dataset, which contains 6 channels. For the univariate datasets (Stock72 and Stock360), the predictive score remains largely unchanged as there are no additional channels to leverage.
>
> We apologize for the confusion caused by the differing evaluation schemes and have clarified this point in the revised manuscript in Sections 4.1 and 4.3. We hope this explanation addresses your concerns and provides a clearer understanding of the results presented in the tables. We have highlighted the expanded discussion in blue in the new manuscript.
>
> **W3.**
> Thank you for highlighting the potential ambiguity in the interpretation of Table 7. We acknowledge that the predictive and discriminative scores may initially seem contradictory. However, they reflect different aspects of model performance.
> The strong predictive scores across all variants indicate their effectiveness in capturing the underlying patterns and dynamics of the time series data useful for forecasting tasks. This suggests that even models with lower discriminative scores, such as iHT (no FFT) and iHT-SIREN, are capable of generating useful predictions.
> Conversely, the lower discriminative scores for these variants reveal that they may struggle to produce samples that are statistically indistinguishable from real data.
> Similarly, in deep learning models for images [3], transformations such as flipping, adding noise, or rotation can enhance model performance (pred.score), even though these altered images remain relatively easy to distinguish from real data (discr.score).
> In Section 4.7 we have expanded the analysis of Table 7, to include the discussion above.
>
> [3] - Shorten, Connor, and Taghi M. Khoshgoftaar. "A survey on image data augmentation for deep learning." Journal of big data 6.1 (2019): 1-48.
>
> **W4.**
> [Update] We have added the additional ablation of Table 7 for irregularly sampled data with its corresponding analysis in Section J.1 in the Appendix.

---

### Review · Reviewer_hq5b · 2024-05-28

**Summary Of Contributions:**

This paper proposes iHyperTime, a framework for generating synthetic time series data. The proposed architecture is based on two key components: TSNet, an Implicit Neural Representation (INR) tailored for encoding time series data in a resolution-agnostic manner, facilitating the disentanglement of trend, seasonality, and residuals within a single time series, and iHyperTime, a hypernetwork architecture that utilizes TSNet to generate interpretable latent representations of time series data, enhancing the generalization capabilities across various types of time series, including multivariate, irregularly sampled, and long sequences. The evaluations demonstrate that iHyperTime outperforms existing state-of-the-art methods in generating synthetic time series, particularly excelling in challenging scenarios such as irregularly sampled or long sequences. The approach achieves rapid training speeds, comparable to the fastest existing methods for short sequences and significantly faster for longer sequences.

**Audience:**

Yes

**Claims And Evidence:**

No

**Requested Changes:**

Rigorous experimentation on irregularly sampled time series data and comaprison with SOTA approaches in this space, e.g. [1]/

**Strengths And Weaknesses:**

### Strengths
- The paper focuses on an important problem of time series generation which can be useful in several practical use cases.
- The paper proposes an interesting idea of disentangling trend, seasonality and residuals, and learning them separately.
- Experiments show that the framework is able to outperform baselines on multiple datasets.
- The paper is easy to follow.


### Weaknesses
- One of the major motivation and focus of the paper seem to revolve around irregularly sampled time series (as mentioned in abstract and introduction), but the paper lacks rigorous experimentation to support the claim that the proposed approach works better on irregularly sampled time series. The paper creates synthetic dataset to replicate `irregular sampling` instead of using widely available irregularly sampled time series datasets such as MIMIC-III, PhysioNet.
- The paper misses some important line of work for irregularly sampled time series and fails to compare with some recent SOTA approaches in this space [1].
- Also the paper claims that the proposed approach outperforms current state-of-the-art methods in challenging scenarios that involve long or irregularly sampled time series, while performing on par on regularly sampled data. It's hard to be convinced that the claim is true since the paper misses SOTA comparison and rigorous experimentation.


References:
[1] S. N. Shukla and B. M. Marlin. Heteroscedastic Temporal Variational Autoencoder for Irregularly Sampled Time Series. In ICLR 2022.

---

> ### Author Response · Authors · 2024-06-12
>
> We appreciate the reviewer's valuable feedback, and the suggested work [1], and datasets (MIMIC-III and PhysioNet).
> In response, we have revised the manuscript to include a comparison between our work and [1] in Appendix A.
> Specifically, we better clarified the challenges of time series generation, which is the focus of our contribution,
> in contrast to tasks such as interpolation or imputation, which are more related to the work in [1].
>
> Time series generation involves creating new data that replicate the patterns and characteristics of the original dataset.
> In contrast, time series interpolation (or imputation) aims to fill in missing values or estimate values at specific time points within an existing series. Although these tasks share some similarities, and methodologies that might inspire one another, they have distinct goals and evaluation metrics.
> Accordingly, we have compared our method (iHyperTime) against a broad range of state-of-the-art approaches and datasets for time-series generation, showing competitive performance. We achieve superior training and sampling times, which we believe is a significant advantage given the challenges, costs, and impact of training deep learning models.
> Our evaluation methodology is consistent to recent work in the time-series literature [2].
>
> Regarding the use of synthetic datasets, we acknowledge the reviewer's suggestion to utilize available irregularly sampled time series datasets such as MIMIC-III and PhysioNet. However, to ensure fair comparisons, we have aligned with the common practices and literature of time series generation: we create irregular data by dropping random values from the time-series [2, 3, 4, 5, 6, 7, 8, 9].
> This controlled approach allows us to effectively evaluate our model's capability to handle varying degrees of irregularity. We have also carefully reviewed the terminology used throughout the manuscript and replaced instances of "missing data" with the more precise term "irregularly sampled" to avoid confusion and more accurately reflect the nature of the data we are working with.
>
> We hope our extensive experiments on both univariate, multivariate, regular, irregular, short, and long time series, across diverse datasets and baselines, provide robust evidence of iHyperTime's effectiveness in generating realistic and useful time series data.
> While we do not claim to have solved the challenges of time series generation, we believe our work demonstrates the potential of using implicit neural representations (INR) for time-series generation, an area that has received limited attention.
> We have added sentences in the conclusion to highlight this potential, the need for further research in this direction, and the need of more work studying and closing the gap between time-series generation and imputation models.
>
> [1] S. N. Shukla and B. M. Marlin. Heteroscedastic Temporal Variational Autoencoder for Irregularly Sampled Time Series. In ICLR 2022.
> [2] Jeon, Jinsung, et al. "GT-GAN: General purpose time series synthesis with generative adversarial networks." Advances in Neural Information Processing Systems 35 (2022): 36999-37010.
> [3] Kidger, Patrick, et al. "Deep signature transforms." Advances in Neural Information Processing Systems 32 (2019).
> [4] Xu, Chen, and Yao Xie. "Conformal prediction interval for dynamic time-series." International Conference on Machine Learning. PMLR, 2021.
> [5] Huang, Zijie, Yizhou Sun, and Wei Wang. "Learning continuous system dynamics from irregularly-sampled partial observations." Advances in Neural Information Processing Systems 33 (2020): 16177-16187.
> [6] Tang, Xianfeng, et al. "Joint modeling of local and global temporal dynamics for multivariate time series forecasting with missing values." Proceedings of the AAAI Conference on Artificial Intelligence. Vol. 34. No. 04. 2020.
> [7] Zhang, Xiang, et al. "Graph-guided network for irregularly sampled multivariate time series." arXiv preprint arXiv:2110.05357 (2021).
> [8] Jhin, Sheo Yon, et al. "Attentive neural controlled differential equations for time-series classification and forecasting." Knowledge and Information Systems 66.3 (2024): 1885-1915.
> [9] Deng, Ruizhi, et al. "Continuous latent process flows." Advances in Neural Information Processing Systems 34 (2021): 5162-5173.

---

### Author Response · Authors · 2024-06-12
**General reply and summary of changes**

We thank the reviewers for their thoughtful and constructive feedback. We are glad that they found our paper to be well-written and easy to follow, and that they appreciated the novelty and effectiveness of our proposed method. We
thank the reviewers for highlighting iHyperTime's strong empirical performance and generalization ability across diverse datasets, as well as its interpretability due to the disentanglement of time series into separate components.

We address the reviewers' concerns and questions in the detailed responses below.
For convenience, we summarize here the main changes we introduced in the manuscript:
- Section 3.1.1: We have added two paragraphs clarifying the relationship between INRs and TSNet.
- Section 4.1: In the Evaluation metrics paragraph, we clarified the difference in the predictive score for regular and irregular time series.
- Section 4.3: We have expanded analysis of results for irregular data.
- Section 4.6: We highlighted additional experiments in the supplementary material that showcase the effectiveness of the decomposition of iHyperTime.
- Section 4.7: We have expanded the analysis of the results for the network ablation.
- Section 5: We have added a paragraph on future work.
- Appendix A: We have added a discussion on related work about time series imputation, highlighting the differences with generation.
- [Update] Appendix J1: We have added additional ablation studies regarding model architecture for irregularly sampled data.

All changes with respect to the previous version have been marked in blue color in the new manuscript.

---

### Decision · Action_Editor_Qc6a · 2024-07-19

**Recommendation:** Accept with minor revision

**Comment:**

I would strongly encourage the authors to address **all** the concerns of the reviewers and particularly the consideration of the related work (and differences from those) and also to strengthen the comparisons with standard datasets, such as MIMIC-III, PhysioNet.

**Audience:**

There are likely individuals interested in the paper, since generation even on timeseries, is relevant to the community.

**Claims And Evidence:**

The paper introduces a hypernetwork and implicit representations for timeseries generation. The claims of the paper are clear to the reviewers and myself. During the review and the discussion phase, there were arguments that the paper uses a synthetic dataset and that the comparisons with SOTA models are not enough. However, the authors did make an effort during the revision to address the concerns for the synthetic dataset, while being SOTA is not a requirement for TMLR acceptance. As such, I do not believe this paper should be rejected.

---

> ### Author Response · Authors · 2024-08-19
>
> We would like to thank again the reviewers and the action editor for their thoughtful and constructive feedback. We appreciate the positive reception of our work and have carefully addressed the remaining comments suggested by the action editor.
>
> Below is a summary of the changes we have made to the manuscript:
> Reviewer 1:
> - Appendix A: A new subsection has been added in related work, discussing time series imputation and highlighting the differences with time series generation.
> - Appendix F: We have included additional comparisons on standard irregularly sampled datasets  (PhysioNet, USHCN), as suggested.
>
> Reviewer 2:
> - Section 4.1: In the Evaluation metrics paragraph, we have clarified the difference in the predictive score for regular and irregular time series.
> - Section 4.3: We have expanded the analysis of results for irregular data.
> - Section 4.7: We have expanded the analysis of the results for the network ablation.
> - Appendix K1: Additional ablation studies focusing on model architecture for irregularly sampled data have been incorporated.
>
> Reviewer 3:
> - Section 3.1.1: We have added two paragraphs clarifying the relationship between INRs and TSNet.
> - Section 4.6: The text now highlights additional experiments presented in the supplementary material, which demonstrate the effectiveness of iHyperTime's decomposition approach.